# Mitotic chromosome binding predicts transcription factor properties in interphase

Mahé Raccaud [1], Elias T. Friman[1], Andrea B. Alber[1], Harsha Agarwal[2], Cédric Deluz[1], Timo Kuhn[2], J. Christof M. Gebhardt [2] & David M. Suter[1]

Mammalian transcription factors (TFs) differ broadly in their nuclear mobility and sequence-specific/non-specific DNA binding. How these properties affect their ability to occupy specific genomic sites and modify the epigenetic landscape is unclear. The association of TFs with mitotic chromosomes observed by fluorescence microscopy is largely mediated by non-specific DNA interactions and differs broadly between TFs. Here we combine quantitative measurements of mitotic chromosome binding (MCB) of 501 TFs, TF mobility measurements by fluorescence recovery after photobleaching, single molecule imaging of DNA binding, and mapping of TF binding and chromatin accessibility. TFs associating to mitotic chromosomes are enriched in DNA-rich compartments in interphase and display slower mobility in interphase and mitosis. Remarkably, MCB correlates with relative TF on-rates and genome-wide specific site occupancy, but not with TF residence times. This suggests that non-specific DNA binding properties of TFs regulate their search efficiency and occupancy of specific genomic sites.

---

[1] Institute of Bioengineering, School of Life Sciences, Ecole Polytechnique Fédérale de Lausanne (EPFL), CH-1015 Lausanne, Switzerland. [2] Institute of Biophysics, Ulm University, Albert-Einstein-Allee 11, 89081 Ulm, Germany. These authors contributed equally: Elias T. Friman, Andrea B. Alber. Correspondence and requests for materials should be addressed to D.M.S. (email: david.suter@epfl.ch)

Transcription factors (TFs) regulate gene expression by binding regulatory sequences of target genes. TF ability to occupy specific genomic sites depends on their nuclear concentration, their ability to search the genome, and the chromatin environment of their binding sites. How TFs maximize search efficiency for specific sites is incompletely understood. Pioneering theoretical work proposed that DNA-binding proteins display substantial non-specific DNA interactions, which modulate TF search efficiency[2]. The length of the DNA sequence flanking the Lac operator was later shown to impact Lac Repressor on-rate, suggesting that local non-specific TF-DNA interactions increase search efficiency by one-dimensional diffusion along DNA[3]. Experimental and computational modeling studies thus converge on a TF search model that combines 3D diffusion and facilitated diffusion, the latter resulting from local 1D search mediated by sliding along DNA, local jumps or hopping, and transfer between genomically-distant but physically close segments of DNA (intersegment transfer)[4–9]. Such local search mechanisms strongly modulate search efficiency and mainly depend on transient non-specific protein-DNA association[1–3,10,11] mediated by electrostatic interactions[12–19]. While gene arrays[20–23] and more recently single molecule imaging[24,25] have allowed monitoring specific DNA-binding events dynamics, non-specific DNA binding of most mammalian TFs remains uncharacterized, and thus to which extent this property impacts genome-wide occupancy of TFs is unknown.

A minority of TFs were shown to associate with mitotic chromosomes[26]. These interactions can be identified by ChIP-seq on mitotic cells and TF-mitotic chromosome co-localization analysis by fluorescence microscopy. While ChIP-seq essentially identifies sequence-specific DNA binding, fluorescence microscopy allows quantifying mitotic chromosome association independently of enrichment on specific genomic sites[26]. Importantly, immunofluorescence protocols involving chemical fixation cause the artifactual eviction of chromatin-bound TFs[27–30]. In contrast, live cell imaging of TFs fused to fluorescent proteins bypass this problem. Both non-specific and specific DNA binding of TFs to mitotic chromosomes have been described. However, the often small number of specifically-bound loci on mitotic chromosomes[31–34], the mild or null sensitivity to alterations of specific DNA binding properties[31,35], and the absence of quantitative relationship between mitotic ChIP-seq datasets and fluorescence microscopy[33] suggest that co-localization of TFs with mitotic chromosomes as observed by microscopy is largely due to non-specific DNA interactions. Converging evidence from the literature further corroborates this view. SOX2 and FOXA1 strongly associate with mitotic chromosomes[31,32] and display high non-specific affinity for DNA in vitro[36,37]. In contrast, OCT4 displays less visible association with mitotic chromosomes[32] and has low non-specific affinity for DNA in vitro[37]. Finally, FOXA1 mutants with decreased non-specific DNA affinity but retaining their specificity for the FOXA1 motif also display reduced mitotic chromosome association[31].

Many TFs binding to mitotic chromosomes have pioneer properties[31,34,38,39], i.e., they can bind and open condensed chromatin regions. However, the existence of a common molecular mechanism underlying mitotic chromosome binding and pioneer activity remains uncertain.

Here we measure mitotic chromosome binding (MCB) of 501 mouse TFs in live mouse embryonic stem (ES) cells. We show that MCB correlates with interphase TF properties such as subnuclear localization, mobility, and with large differences in TF ability to occupy specific genomic sites. We propose that the co-localization of TFs with mitotic chromosomes is a proxy for TF non-specific DNA binding properties, which regulate TF search efficiency for their specific binding sites and thereby their impact on chromatin accessibility.

## Results

**Large-scale assessment of TF binding to mitotic chromosomes.** To measure MCB for a large number of TFs, we constructed a doxycycline (dox)-inducible lentiviral vector library of 757 mouse TFs fused to a yellow fluorescent protein (YPet) (Fig. 1a). This library was used to generate a corresponding library of mouse embryonic stem (ES) cell lines to quantify TF MCB by live cell fluorescence microscopy. Cells were seeded in 96-well plates and treated with dox to induce expression of TF-YPet fusion proteins, and the next day they were imaged by wide-field fluorescence microscopy. We used a semi-automated pipeline to detect cells in metaphase, allowing easy quantification of MCB since chromosomes are most spatially confined in this phase. Of note, we did not observe obvious differences in TF co-localization with mitotic chromosomes between prophase, metaphase and anaphase. We used the Mitotic Bound Fraction (MBF) as a metric for mitotic chromosome binding, defined as the averaged YPet fluorescence intensity on metaphase chromosomes multiplied by the fraction of cellular volume occupied by DNA (as measured by confocal microscopy, see Methods), divided by the total YPet signal (Fig. 1a). We then asked whether MBF values depend on over-expression levels, in at least 19 individual cells for each of 21 different TFs spanning a broad range of MBF. We found no consistent correlation between these two parameters (Supplementary Fig. 1a), suggesting that the MBF is largely independent of overexpression levels. We also measured the MBF of an endogenously expressed TF and the MBF of its overexpressed counterpart, using a homozygous SOX2-SNAP cell line[40] and a dox-inducible SOX2-SNAP cell line[41]. We only observed modest differences between their MBF values (Supplementary Fig. 1b). We then compared wide-field fluorescence measurements with those performed by confocal microscopy, and found these to be in very good agreement (Supplementary Fig. 1c-d, Supplementary Table 1). In total 501 TFs yielded sufficiently strong fluorescent signals in metaphase to allow measuring their MBF, and for 94% of these we could measure the MBF in at least 10 cells (see Methods). We defined three bins of TFs based on visual inspection of the YPet signal on metaphase chromosomes: depleted (YPet signal lower than in the cytoplasm), intermediate (YPet signal equal to that of the cytoplasm) or enriched (YPet signal higher than in the cytoplasm), corresponding to MBFs <16.5%, 16.5–23% and >23%, respectively. Twenty-four percent of TFs fell in the depleted bin, 54% in the intermediate bin, and 22% in the enriched bin (Fig. 1b). Most TFs previously reported as enriched on mitotic chromosomes and for which we could obtain MBF values, such as FOXA1[31], GATA1[34], SOX2[27,32], ESRRB[35], HMGB2[29,31], and HMGN1[29], fell in the intermediate (ESRRB) or enriched (all five other TFs) category (Supplementary Table 2), suggesting that C-terminal YPet fusion does generally not perturb MCB. We also measured the MBF of a subset of TFs in NIH-3T3 cells, yielding consistent results (Supplementary Fig. 1e-f, Supplementary Table 3), suggesting that intrinsic TF properties are major determinants of mitotic chromosome association. We next compared our results with those obtained by a proteomics study investigating chromatin-bound proteins over the cell cycle[42]. Importantly, that study used a different cell type (a glioblastoma cell line), species (human), and involved PFA cross-linking that can lead to artifactual disruption of TF-mitotic chromosome interactions[27–30]. Nevertheless, we found our results to globally agree with theirs (Supplementary Fig. 1g). Furthermore, while in contrast to the study by Ginno et al.[42], our library contains TFs

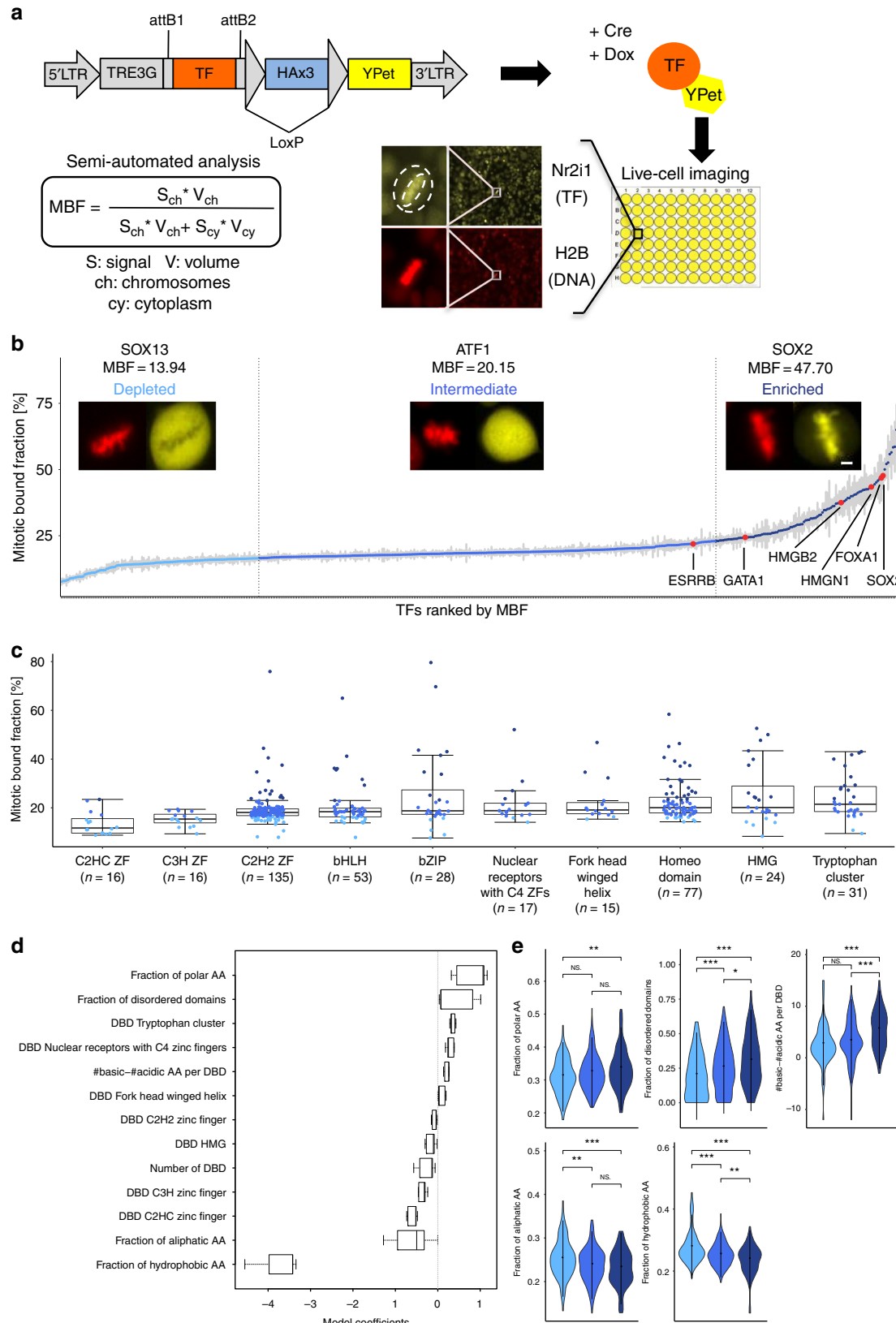

that are endogenously expressed in ES or NIH-3T3 cells but also many that are not, we did not find significant differences in MBF values between these two groups (Supplementary Fig. 1h). This suggests that endogenous TF expression status does not strongly impact its MBF.

Next, we clustered TFs according to their DNA binding domains (DBD) and found that members of some TF families (e.g., homeodomain or tryptophan cluster) were more likely to be enriched on mitotic chromosomes (Fig. 1c). In contrast, C2H2 Zinc-finger TFs were underrepresented in the enriched category,

**Fig. 1** Large scale quantification of mitotic chromosome binding of TFs. **a** Experimental strategies for generating a lentiviral and ES cell TF-YPet expression library and for quantifying their association to mitotic chromosomes. **b** Mitotic bound fraction of 501 transcription factors, ranked from lowest to highest and grouped in three bins according to the color code described in the caption (for *n* values see Supplementary Table 2). Microscopy inset show representative images of TF localization in metaphase for each category. Scale bar: 3 μm. Error bars: SEM. **c** Mitotic bound fraction of TFs with different types of DNA binding domains. The color code is the same as in panel **b**. Boxes: intervals between the 25th and 75th percentile and median (horizontal line). Error bars: 1.5-fold the interquartile range or the closest data point when no data point is outside this range. **d** Parameters recovered by machine learning that impact the MBF and retained in the model for >90% of the runs (*n* = 500). Boxes: intervals between the 25th and 75th percentile and median (horizontal line). Error bars: 1.5-fold the interquartile range or the closest data point when no data point is outside this range. **e** Violin plots of TF distributions for the fraction of polar amino acids, fraction of disordered domains, number of basic residues minus the number of acidic residues, dispersion of positive charges, and fraction of hydrophobic amino acids, grouped in the same categories as in panel **b** ($n_{\text{Depleted}}$ = 118, $n_{\text{Intermediate}}$ = 272, $n_{\text{Enriched}}$ = 111). ∗*p*-value<0.05; ∗∗*p*-value<0.01; ∗∗∗*p*-value<0.001; NS not significant (Wilcoxon rank-sum test). *P*-values were obtained using a Wilcoxon rank-sum test

in line with mitotic phosphorylation preventing their association with DNA during M-phase[43,44]. However, the broad range of MBF within each family indicates that DNA binding domain type does not strictly govern the MBF (Fig. 1c), suggesting other TF characteristics involved in regulating mitotic chromosome association.

**The MBF depends on electrostatic properties**. We next used a machine learning algorithm (see Methods) based on a lasso regularized generalized linear model[45] to uncover TF features that could explain differences in MBF (Fig. 1d and Supplementary Fig. 1i). We collected a large number of features from the amino acid sequence of 401 different TFs (see Methods and Supplementary Table 4) to find those that are correlated with the MBF. We ran the algorithm 500 times on the data to obtain parameters used to predict the MBF of the remaining 100 TFs. The algorithm selects variables allowing predicting the MBF, while coefficients of non-predictive parameters are set to zero. Similarly, coefficients of covariates that are correlated among them are set to zero to select a reduced subset of predictive variables. We then kept only parameters with a coefficient >0 in at least 90% of the runs (Fig. 1d). As expected, certain types of DBD, such as the tryptophan cluster TF family and the C2HC Zinc fingers were moderately predictive of the MBF. The other parameters most strongly correlated with the MBF were the fractions of polar amino acids and disordered domains, and the absolute charge per DBD (number of basic minus number of acidic residues), while the fraction of aliphatic and hydrophobic amino acids was negatively correlated with the MBF (Fig. 1d). We then compared these parameters between the three different TF bins we defined above (Fig. 1b). Although broadly distributed in each bin, all of them were significantly different in TFs enriched on mitotic chromosomes (Fig. 1e). The absolute charge per DBD was the most distinctive parameter between TFs enriched on mitotic chromosomes (dark blue) versus those that are not (medium and light blue), suggesting that electrostatic interactions play an important role in mitotic chromosome association of TFs.

TFs usually harbor one or several nuclear localization signals (NLS), which often consist of a series of positively charged amino acids. During mitosis, Ran guanine nucleotide exchange factor (RCC1) associates with mitotic chromosomes, thereby maintaining a GTP/GDP gradient between mitotic chromosomes and the cytoplasm[46]. As NLS sequences were suggested to mediate MCB through active transport[27,28], we aimed to determine whether the impact of positive charges could be confounded by active NLS-mediated transport of TFs to mitotic chromosomes. To do so, we engineered YPet fusions to either an NLS or an equivalent number of positive charges not predicted to mediate nuclear import. The addition of positive charges to the YPet protein was sufficient to increase its co-localization with mitotic chromosomes, independently of nuclear import (Supplementary Fig. 1j).

We also added five positively charged amino acids to the N-terminus of four different TFs, and as expected these displayed an increased MBF (Supplementary Fig. 1k). Taken together, these results are in line with previous studies suggesting that (i) non-specific DNA binding is essentially mediated by electrostatic protein-DNA interactions, and (ii) non-specific DNA binding is largely responsible for the co-localization of TFs with mitotic chromosomes.

**The MBF correlates with interphase TF-DNA co-localization**. Since mitotic chromosome association reflects general non-specific DNA binding properties, we next asked whether the MBF scales with TF-DNA co-localization in interphase. To address this question, we used NIH-3T3 cells, which display easily identifiable regions of varying chromatin densities. We stained NIH-3T3 cells with Hoechst and classified the signal in three bins using automatic image thresholding: (i) very dense, H3K9me3-enriched heterochromatin regions (Fig. 2a); (ii) DNA-rich regions; iii) DNA-poor regions (Fig. 2b, Methods). We then generated 38 NIH-3T3 cell lines allowing dox-inducible expression of selected TF-YPet fusions spanning a broad range of MBF. After overnight dox treatment and Hoechst staining, we performed two-color confocal microscopy to measure YPet-Hoechst signal co-localization (Fig. 2c). We observed a wide range of spatial repartition of TFs, with some of them (e.g., Duxbl, Bhlhb8, and Dlx6, Fig. 2c) highly enriched in heterochromatic regions. The MBF and the co-localization of TFs with DNA in interphase were positively correlated (Fig. 2d and Supplementary Table 5). We then determined this correlation within the different sub-nuclear regions we defined, and found the MBF to be positively correlated with enrichment in heterochromatin regions (Fig. 2e and Supplementary Table 5) and to a lesser extent with enrichment in DNA-rich regions (Fig. 2f and Supplementary Table 5). In contrast, the MBF was strongly, inversely correlated with enrichment in DNA-poor regions (Fig. 2g and Supplementary Table 5). Thus, TFs with a high MBF tend to be excluded from DNA-poor regions and are distributed mainly within hetero-chromatic and DNA-rich regions in interphase, suggesting that non-specific TF-DNA interactions also control interphase TF localization.

**The MBF correlates with mitotic and interphase TF mobility**. We reasoned that co-localization with DNA-rich regions resulting from transient TF-DNA interactions should decrease TF mobility in the cell. To test this hypothesis, we performed fluorescence recovery after photobleaching (FRAP) experiments in ES cells on 15 selected TFs with different MBF in mitotic and interphase cells, as well as FRAP on interphase cells only for 3 TFs excluded from mitotic chromosomes (Fig. 3a, b, Supplementary Table 6). We selected these 18 TFs based on their high signal to noise ratio required for reliable FRAP measurements,

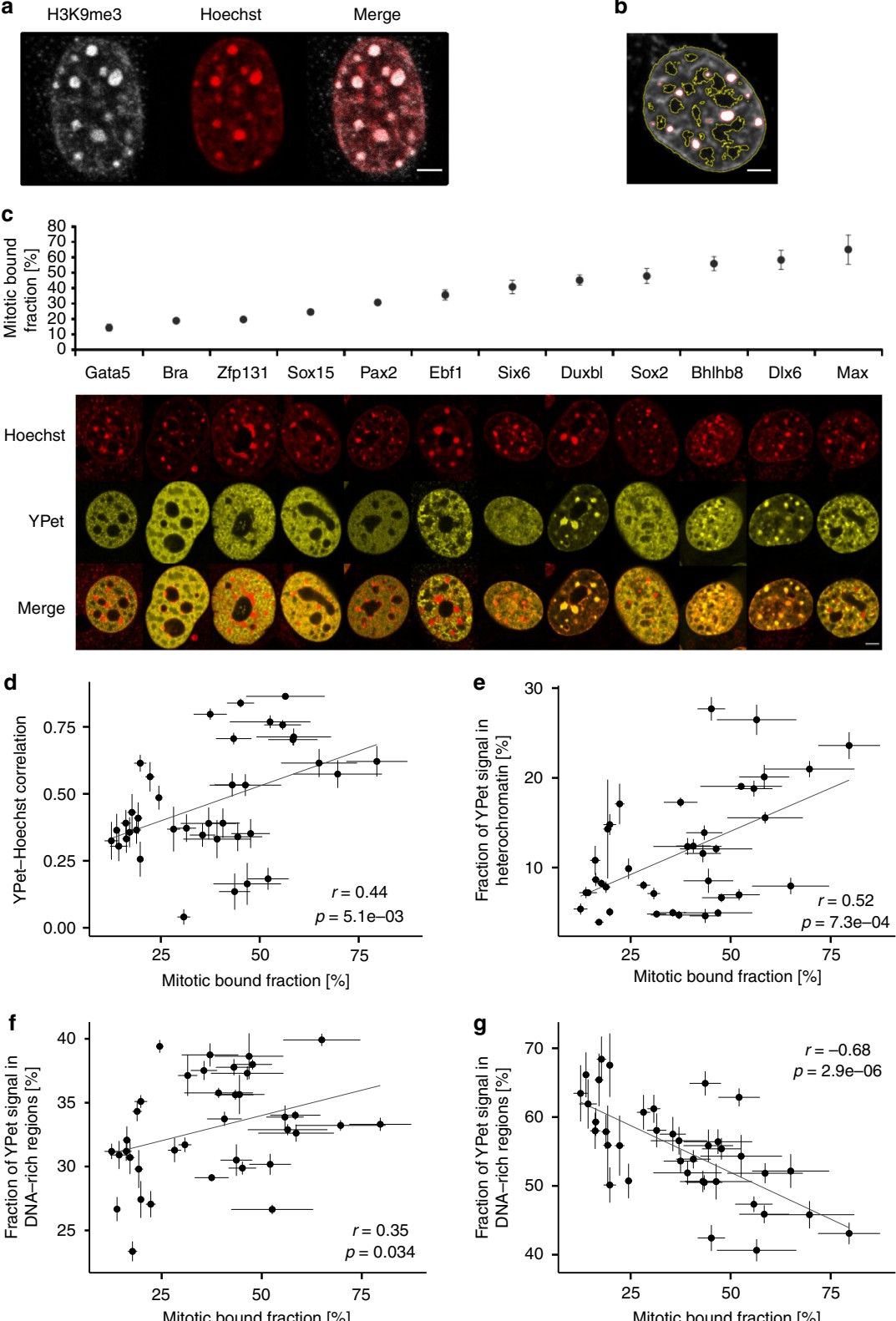

**Fig. 2** Mitotic chromosome binding is correlated with TF-DNA co-localization in interphase. **a** Immunofluorescence labeling of H3K9me3 (Abcam, #ab8898) and Hoechst staining of a NIH-3T3 nucleus. Scale bar: 3 μm. **b** Automatic detection of regions displaying different densities of DNA: Heterochromatic (circled in red), DNA-rich (between red and yellow circles), or DNA-poor (circled in yellow). Scale bar: 3 μm. **c** Examples of TF-YPet interphase localization in NIH-3T3 cells, as compared to Hoechst staining and ranked by mitotic bound fraction (for *n* values see Supplementary Table 2). Bra Brachyury. Scale bar: 5 μm. **d** Correlation between the TF-YPet/Hoechst co-localization (MAFK: *n* = 8, NANOG and BHLHB8: *n* = 11, others: *n* = 10) and the mitotic bound fraction. **e**-**g** Correlation between the mitotic bound fraction and the fraction of YPet signal co-localized with heterochromatic regions (**e**), DNA-rich regions (**f**), and DNA-poor regions (**g**). *n* = 10. Error bars: SEM. *r*-values (*r*) and *p*-values (*p*) are based on Pearson correlation

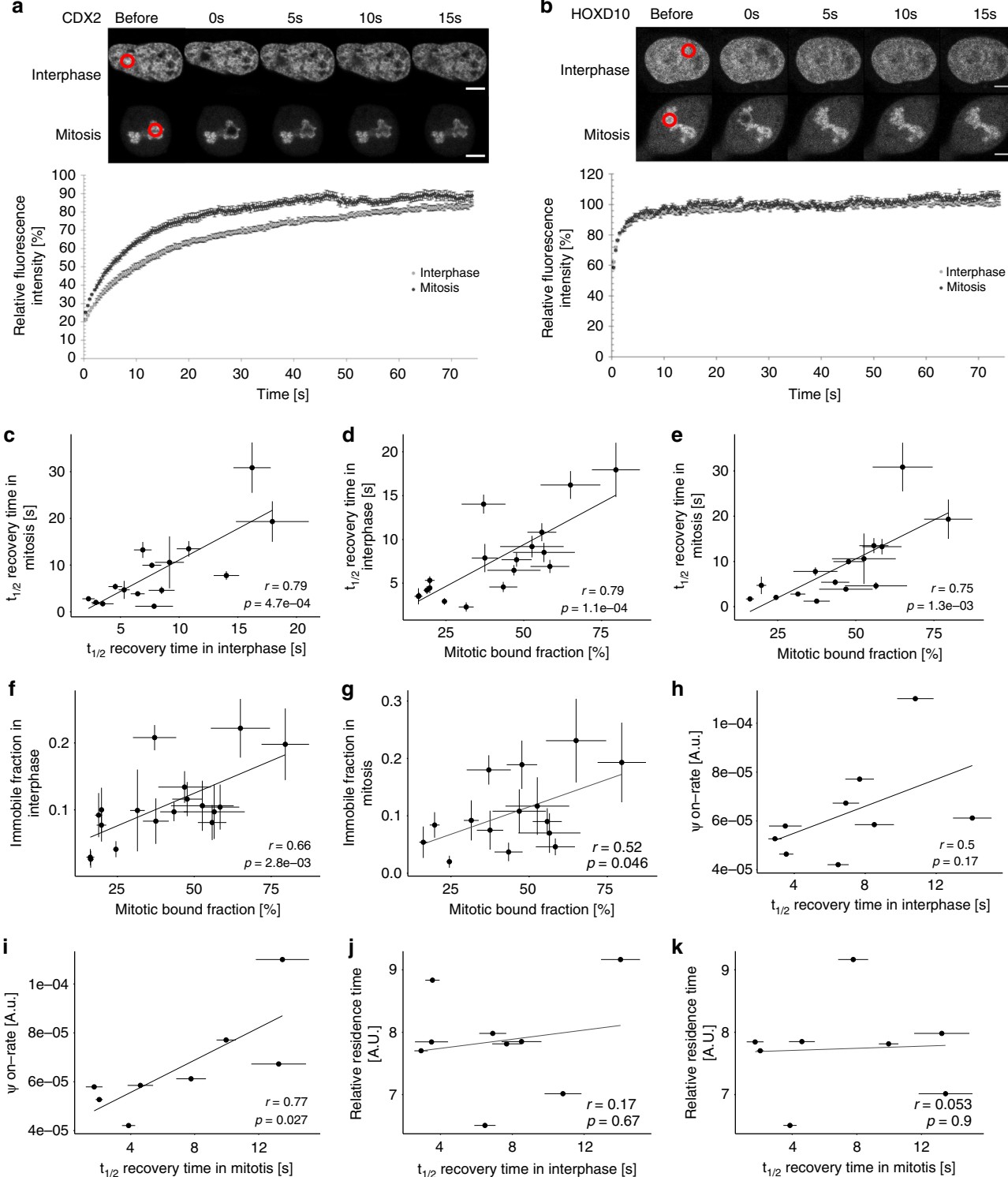

**Fig. 3** TF mobility correlates with the mitotic bound fraction and interphase ψon-rate. **a**, **b** Top: Example of FRAP for CDX2-YPet (**a**) and HOXD10-YPet (**b**) for cells in interphase and mitosis (scale bar: 5 μm). Bottom: normalized interphase and mitosis FRAP recovery curves for CDX2-YPet (**a**) and HOXD10-YPet (**b**). **c** Correlation between FRAP $t_{1/2}$ recovery of TFs in interphase and mitosis. **d** Correlation between TF mitotic bound fraction and FRAP $t_{1/2}$ recovery in interphase. **e** Correlation between TF mitotic bound fraction and FRAP $t_{1/2}$ recovery in mitosis. **f** Correlation between TF mitotic bound fraction and FRAP immobile fraction in interphase. **g** Correlation between TF mitotic bound fraction and FRAP immobile fraction in mitosis. **h** Correlation between FRAP $t_{1/2}$ recovery in interphase and the ψon-rate. **i** Correlation between FRAP $t_{1/2}$ recovery in mitosis and the ψon-rate. **j** Correlation between FRAP $t_{1/2}$ recovery in interphase and TF relative residence times. **k** Correlation between FRAP $t_{1/2}$ recovery in mitosis and TF relative residence times. Error bars: SEM. *r*-values (*r*) and *p*-values (*p*) are based on Pearson correlation. For FRAP relative intensity, $t_{1/2}$ recovery time and immobile fraction: $n = 10$

and their coverage of a broad range of MBF. We then used half-time ($t_{1/2}$) of fluorescence recovery as a metric of TF mobility. Mitotic and interphase $t_{1/2}$ of fluorescence recovery displayed a strong positive correlation (Fig. 3c and Supplementary Table 6), and TF mobility was strongly correlated with the MBF (Fig. 3d, e). These results suggest that intrinsic TF properties govern the MBF and TF mobility in both interphase and mitosis.

**TF size or off-rate do not explain differences in mobility**. FRAP recovery times depend on 3D diffusion, specific and non-specific DNA binding[47]. Since mitotic chromosome association observed by fluorescence microscopy largely depends on non-specific TF-DNA interactions and correlates with $t_{1/2}$ of FRAP, we reasoned that differences in TF mobility might be mainly due to differences in non-specific DNA binding. However, since 3D diffusion and specific DNA binding also influence $t_{1/2}$ of fluorescence recovery, we first asked whether differences in these parameters could explain differences in TF mobility. According to the Stokes-Einstein equation, diffusion scales inversely with the size of molecules. Even though differences in TF radius are predicted to be very small since they scale with the third root of their mass, we assessed the correlation between TF-YPet molecular weight and FRAP $t_{1/2}$ recovery (Supplementary Fig. 2a, b and Supplementary Table 6). These two parameters were negatively correlated and thus do not explain differences in TF mobility. To test whether TFs quantitatively differ in their association with specific DNA sites, we determined the immobile fraction, which corresponds to the fraction of molecules that do not exchange over the time course of FRAP experiments. The immobile fractions in interphase and mitosis were well correlated with the mitotic bound fraction (Fig. 3f, g), suggesting that TFs with a high MBF tend to have a higher fraction of long binding events.

We then performed single molecule (SM) imaging of TFs in live cells to monitor specific DNA binding events. This approach allows comparing frequencies of specific DNA-binding events between TFs at a given TF nuclear concentration, thus providing a relative measure of TF on-rates. Furthermore, SM imaging allows determining differences in TF residence times on specific DNA sites. We generated 9 NIH-3T3 cell lines allowing dox-inducible expression of TFs C-terminally fused to a HaloTag and induced TF expression with low doses of dox shortly before performing SM imaging in interphase cells, using highly inclined and laminated optical sheet (HILO) microscopy[24] (see Methods). Single molecules were registered as bound to chromatin when they were confined for several frames inside a specified region (see Methods). We then associated these binding events with areas of bright, intermediate or dark Hoechst staining intensity (Supplementary Fig. 3a and Supplementary Table 7), similarly as for confocal measurements of TF nuclear distribution (see Fig. 2a, b). In order to compare different TFs, we calculated the ratio of DNA-binding events divided by the total number of detected molecules, the area of the corresponding intensity class and the recording time (Eq. 7 in Methods). This ratio provides a frequency of DNA binding events. We referred to this frequency for binding events lasting >1 s as the pseudo on-rate ($\psi$on-rate hereafter). We approximated the residence time of a TF with the average time a molecule spent bound to DNA (Equation 8 in Methods). FRAP $t_{1/2}$ recovery times in interphase and mitosis were correlated with interphase TF $\psi$on-rates (Fig. 3h, i, Supplementary Tables 6 and 7). In contrast, FRAP recovery times were not correlated with relative interphase residence times (Fig. 3j, k, Supplementary Tables 6 and 7). This suggests that while differences in TF $\psi$on-rates could contribute to differences in TF mobility, differences in 3D diffusion or residence times on specific DNA sites do not. This is also in line with non-specific

DNA binding contributing to differences in TF mobility and search efficiency, thereby impacting TF $\psi$on-rates on specific sites.

**TF occupancy correlates with $\psi$on-rates but not off-rates**. We next aimed to determine the genome-wide TF occupancy at specific sites for 21 TFs spanning a broad range of MBF, and we also included two FOXA1 mutants that have reduced electrostatic interactions with DNA, decreasing non-specific DNA binding activity without altering their specificity for the FOXA1 DNA motif[31]. As expected, FOXA1 mutants displayed a lower MBF than wild-type FOXA1[31] (Supplementary Fig. 3b). For each TF, we generated an NIH-3T3 cell line allowing their dox-inducible expression with three HA tags fused to their C-terminus. We verified that different TFs were expressed at roughly comparable levels upon dox induction (Supplementary Fig. 3c, Supplementary Table 8), allowing ChIP-seq measurements to provide relative measurements of genome occupancy within a given TF concentration range. We then performed ChIP-seq using the same anti-HA antibody for all TFs and used peak calling to determine the number of specifically bound sites. Remarkably, for the 8 TFs for which we obtained both ChIP-seq and single molecule measurements, TF $\psi$on-rates were strongly correlated with the number of ChIP-seq peaks (Fig. 4a, Supplementary Fig. 3d-f, Supplementary Tables 7 and 8). The offset of the linear fit suggests that a fraction of long-lived SM events is not detected by ChIP-seq. As previously reported, these can result from long-lived, non-specific DNA binding events or indirect TF-DNA interactions[23]. In contrast, the number of ChIP-seq peaks was not correlated with TF relative residence times (Fig. 4b, Supplementary Tables 7 and 8). This suggests that the differences in genome-wide occupancy of TFs we observed are mainly due to differences in their association rate but not to longer residence times on specific DNA sites.

**The MBF correlates with genome-wide TF occupancy**. We used the ChIP-seq data to compare relative TF occupancy of specific sites in the genome with their MBF. We first determined the number of ChIP-seq peaks obtained for the 21 TFs, the two FOXA1 mutants, with two biological replicates for OCT4, SOX2, FOXA1, and BHLHB8 to estimate variability in recovering ChIP-seq peaks. Remarkably, the MBF and the number of ChIP-seq peaks were strongly correlated, with differences in ChIP-seq peak numbers ranging over three orders of magnitude (Fig. 4c, d and Supplementary Table 8). This correlation was robust to different $q$-value thresholds (Supplementary Fig. 4a), peak calling algorithms (Supplementary Fig. 4b), and downsampling to equalize the number of reads for each TF (Supplementary Fig. 4c). Very similar results were obtained when using the fraction of ChIP-seq reads in peaks as another metric for TF occupancy (Supplementary Fig. 4d and Supplementary Table 8), indicating that differences in ChIP-seq peak amplitude are small compared to differences in peak numbers. According to the law of mass action, the rate of formation of a TF-DNA complex scales linearly with its concentration, and therefore the six-fold range in TF expression level (Supplementary Fig. 3c) is unlikely to account for the very large differences in ChIP-seq peak numbers we observed. This relationship was also independent of the presence or absence of endogenous expression of these TFs in NIH-3T3 cells (Fig. 4d and Supplementary Table 8).

We then wondered why some TFs reported to display a high number of ChIP-seq peaks when expressed in their endogenous context displayed orders of magnitude fewer peaks in NIH-3T3 cells. While this may partly be due to differences in TF concentrations, antibodies used, and ChIP-seq protocols, we

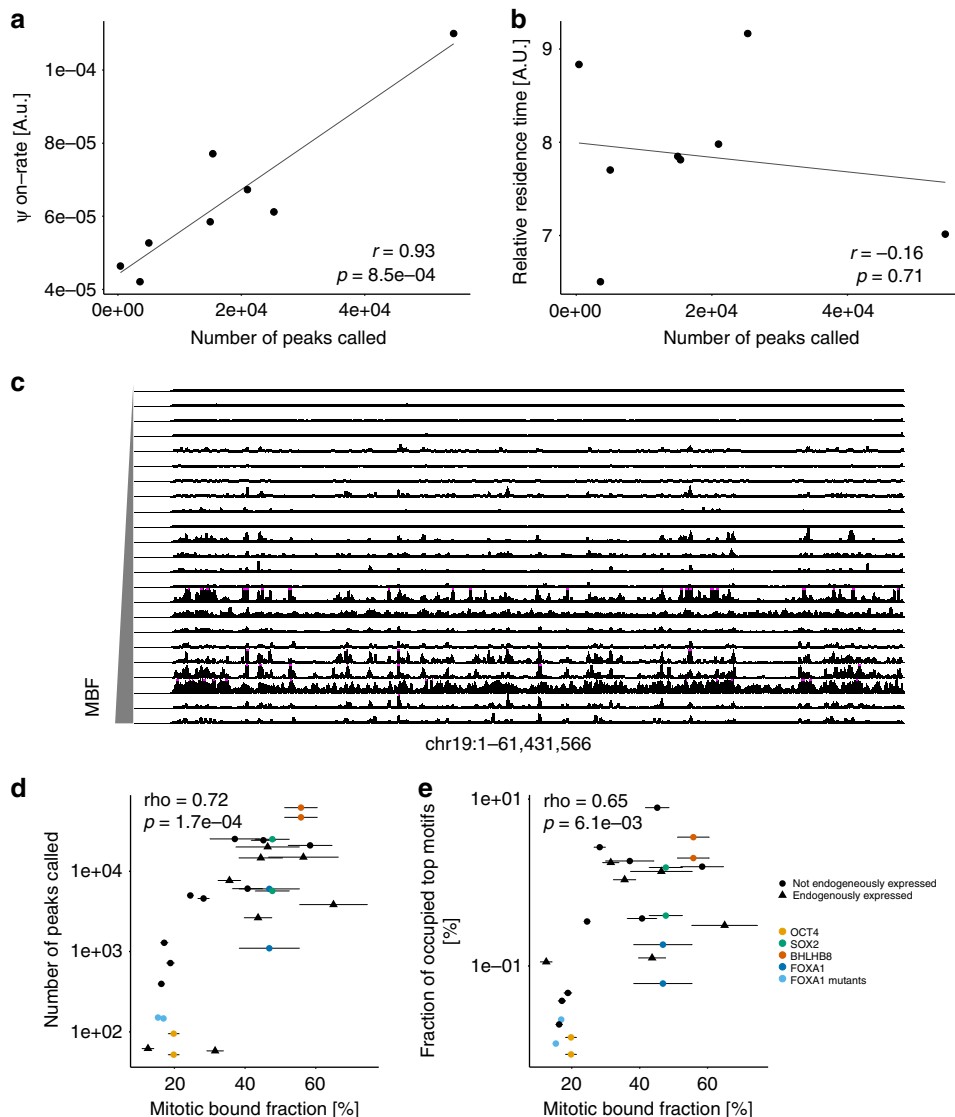

**Fig. 4** Mitotic chromosome binding predicts genome-wide TF occupancy. **a** Correlation between the number of ChIP-seq peaks called and the ψon-rate for 8 different TFs. r-value (r) and p-value (p) are based on Pearson correlation. **b** Correlation between the number of ChIP-seq peaks called and TF relative residence times for 8 different TFs. r-value (r) and p-value (p) are based on Pearson correlation. **c** Genome tracks of RPKM-normalized bigWig files for each TF. The region encompasses the entire chromosome 19. The y-axis is scaled equally for all factors. **d** Correlation between the mitotic bound fraction and the number of ChIP-seq peaks called. Duplicates are indicated for OCT4 (yellow), SOX2 (green), BHLHB8 (red), and FOXA1 (dark blue). The two FOXA1 mutants are shown in light blue. Triangles TFs endogenously expressed in NIH-3T3. Circles: TFs not endogenously expressed in NIH-3T3. Rho-value (Rho) and p-value (p) are based on Spearman's rank correlation. **e** Correlation between the mitotic bound fraction and the number of ChIP-seq peaks displaying the most frequently found motif for each TF, normalized over the total number of motif occurrences in the genome. Same color and shape coding as **d**. Rho-value (Rho) and p-value (p) are based on Spearman's rank correlation. Error bars: SEM

reasoned that in some cases differences in binding partners could also explain this discrepancy. In particular, OCT4 displayed two orders of magnitude fewer ChIP-seq peaks in NIH-3T3 as compared to ES cells[48]. As SOX2 and OCT4 form heterodimers, and SOX2 can recruit OCT4 to DNA[49,50], OCT4 may have a low intrinsic ability to find its target sites in the absence of a heterodimeric partner. To test this hypothesis, we co-expressed a YPet-SOX2 fusion protein with OCT4-HA in NIH-3T3 and performed ChIP-seq against OCT4. While co-expression of YPet-SOX2 did not alter the expression level of OCT4-HA (Supplementary Fig. 4e), the number of peaks (Supplementary Fig. 4f) and fraction of reads in peaks strongly increased (Supplementary Fig. 4g) of OCT4-HA, suggesting that SOX2 drives OCT4 DNA binding in NIH-3T3 cells.

**The MBF scales with TF search efficiency for specific sites**. The number of sites bound by each TF may also depend on the number of specific sites available in the genome. We thus quantified the number of occurrences of the most highly enriched (top) motif for the bound sites of each TF (see Supplementary Table 9 and Methods). We then quantified the fraction of each top motif that is occupied by its respective TF by dividing the number of ChIP-seq peaks containing the top motif (Supplementary Tables 8 and 9) by the total number of its occurrences in the genome (fraction of occupied motifs or FOM). In addition, we performed ATAC-seq on NIH-3T3 cells to map accessible chromatin regions, allowing to calculate the FOM in open chromatin regions. The FOM in both the whole (Fig. 4e) and accessible genome (Supplementary Fig. 4h) strongly correlated

with the MBF (Supplementary Table 8), suggesting large differences in TF ability to find their specific sites, independently of the number of sites present in the whole genome or within accessible chromatin regions. Notably, some TFs were not enriched for their known consensus motif from the literature but rather for motifs annotated to other factors. This is not unexpected since TF binding profiles are often context-dependent[51,52], and motif enrichment can also be due to indirect binding or the presence of a motif in proximity of a true binding site. To exclude the possibility that our correlation is biased by indirect binding or incorrect motif identification, we analyzed the FOM at known consensus sites for all factors significantly enriched for a motif annotated to the factor itself or a closely related protein (e.g., the SOX3 motif for SOX2 and SOX15, see Supplementary Table 9 and Methods) and found this metric to also positively correlate with the MBF (Supplementary Fig. 4i, j and Supplementary Table 8). Therefore, our data suggests that the MBF is of strong predictive value for the ability of TFs to occupy their specific sites in the genome at a given TF concentration.

**MCB does not predict pioneer activity**. We then asked whether the MBF is predictive of TF ability to modify chromatin accessibility at their binding sites. Since pioneer TFs are suggested to be more capable of targeting closed chromatin regions than non-pioneer TFs[53], we first determined if the MBF was correlated with a higher propensity to bind closed chromatin regions. We found no correlation between the MBF and the fraction of TF ChIP-seq peaks in closed chromatin (Fig. 5a and Supplementary Table 8), and a weak positive correlation between the MBF and the FOM in closed chromatin (Supplementary Fig. 5a). We then selected 13 TFs that are not endogenously expressed in NIH-3T3 cells based on a published RNA-seq dataset[54] to interrogate their ability to modify chromatin accessibility. For each TF, we performed two biological replicates of ATAC-seq after 48 h of dox induction in the respective NIH-3T3 cell line (Fig. 5b and Supplementary Fig. 5b). We counted the number of ATAC-seq reads in regions bound by each TF in the control cell line and each overexpression cell line and used edgeR and limma to call regions significantly ($p < 0.05$) either more or less accessible following overexpression of the TF (see Methods). Overall, the number of regions with changed accessibility coinciding with TF ChIP-seq peaks correlated with the number of TF ChIP-seq peaks (Fig. 5c and Supplementary Table 10). Of note, these regions displayed either significantly increased (Supplementary Fig. 5c) or decreased (Supplementary Fig. 5d) accessibility, and both correlated with the number of TF ChIP-seq peaks. As expected, a large number of genomic sites to which known pioneer TFs such as SOX2 and FOXA1 bound displayed changes in chromatin accessibility (Fig. 5c, Supplementary Fig. 5b, and Supplementary Table 10). The impact of TFs on both opening and closing chromatin can be explained by the fact that many of these TFs were reported to function as both activators and repressors[48,55–60]. Importantly, these correlations were maintained when analyzing the same regions for all TFs (defined as all regions with an ATAC-seq peak in at least one sample, see Methods) (Supplementary Fig. 5e-g). To determine the intrinsic ability of each TF to modify chromatin accessibility, we quantified the fraction of sites bound by each TF that displayed significant increase or decrease in chromatin accessibility. Neither of them was correlated with the MBF (Fig. 5d-e and Supplementary Table 10), suggesting that TFs with a high MBF are not more potent on average in altering chromatin accessibility. Therefore, a high MBF is not a signature for an intrinsically higher propensity to alter chromatin accessibility.

## Discussion

Recent studies have shown that TFs binding to mitotic chromosomes are rather common[26,61], and these were proposed to play a role in cell fate maintenance by controlling gene reactivation early during mitotic exit[31,34,39,62]. Furthermore, the presence of SOX2 and OCT4 at the M-G1 transition was shown to be required for their role in regulating ES cell fate decisions[32,63]. However, the molecular mechanisms underlying MCB are largely unknown and this property remains uncharacterized for the vast majority of TFs. Here we measured MCB of 501 TFs and found over 100 to be enriched on mitotic chromosomes. Our study thus provides a large database of TFs to be mined for their potential role in cell fate maintenance during cell division.

MCB as observed by fluorescence microscopy is distinct from the notion of mitotic bookmarking, which implies TF binding to specific sites during mitosis. However, all mitotic bookmarking TFs described so far are also decorating mitotic chromosomes as assessed by fluorescence microscopy[27,31,32,34,35,62,63]. Here we show that TFs with a high MBF have a higher immobile fraction on mitotic chromosomes, raising the possibility that these also display a higher occupancy of specific sites during mitosis. However, mitotic ChIP-seq results obtained by different laboratories show little agreement[32,33,63], and the scarcity of data on TFs displaying poor co-localization with mitotic chromosomes by fluorescence microscopy does not allow to draw clear conclusions on the link between these two properties.

While our data indicate that the type of DNA binding domain and electrostatic characteristics of TFs impact MCB of TFs, the in silico determination of these properties based on TF amino acid sequence was not sufficient to accurately predict the MBF. Therefore, measuring other parameters such as post-translational modifications and three-dimensional structures of TF-DNA contact interfaces is arguably required to improve MBF prediction.

We found that quantitative measurements of MCB are well correlated to other TF properties. First, differences in the MBF scaled inversely with TF mobility in both interphase and mitosis. This can be explained by differences in non-specific DNA binding properties, which drive mitotic chromosome association and decrease TF mobility in the nucleus. Second, less mobile TFs displayed higher ψon-rates, suggesting that non-specific DNA association increases TF efficiency to search for target sites through facilitated diffusion. This is also corroborated by the high and low genome occupancy of TFs (SOX2 or FOXA1 versus OCT4) with high and low non-specific DNA binding activity in vitro, respectively[37]. Third, TFs differed broadly in their ability to occupy specific sites in the genome during interphase, which scaled with their MCB, even though these two measurements were performed in different cell types. Since most TFs we studied are not endogenously expressed in NIH-3T3 cells, they are likely to depend essentially on their intrinsic ability to find their specific sites in the genome rather than on the cooperativity with other TFs. Importantly, here we aimed to determine the intrinsic ability of TFs to bind to their sites within a given concentration range. Thus, optimization of TF expression levels in their physiological context may further allow fine-tuning their genome occupancy. Interestingly, we found that OCT4 occupancy can be dramatically increased by co-expression of SOX2, and we had reported that SOX2 expression increases mitotic chromosome association of OCT4[32]. Therefore, it is possible that some TFs depend on cooperativity with other DNA-binding proteins to mediate or enhance MCB and search for specific binding sites.

Finally, our data also sheds light on the proposed link between MCB and impact on chromatin accessibility. The MBF could predict neither preferential binding to closed chromatin nor the

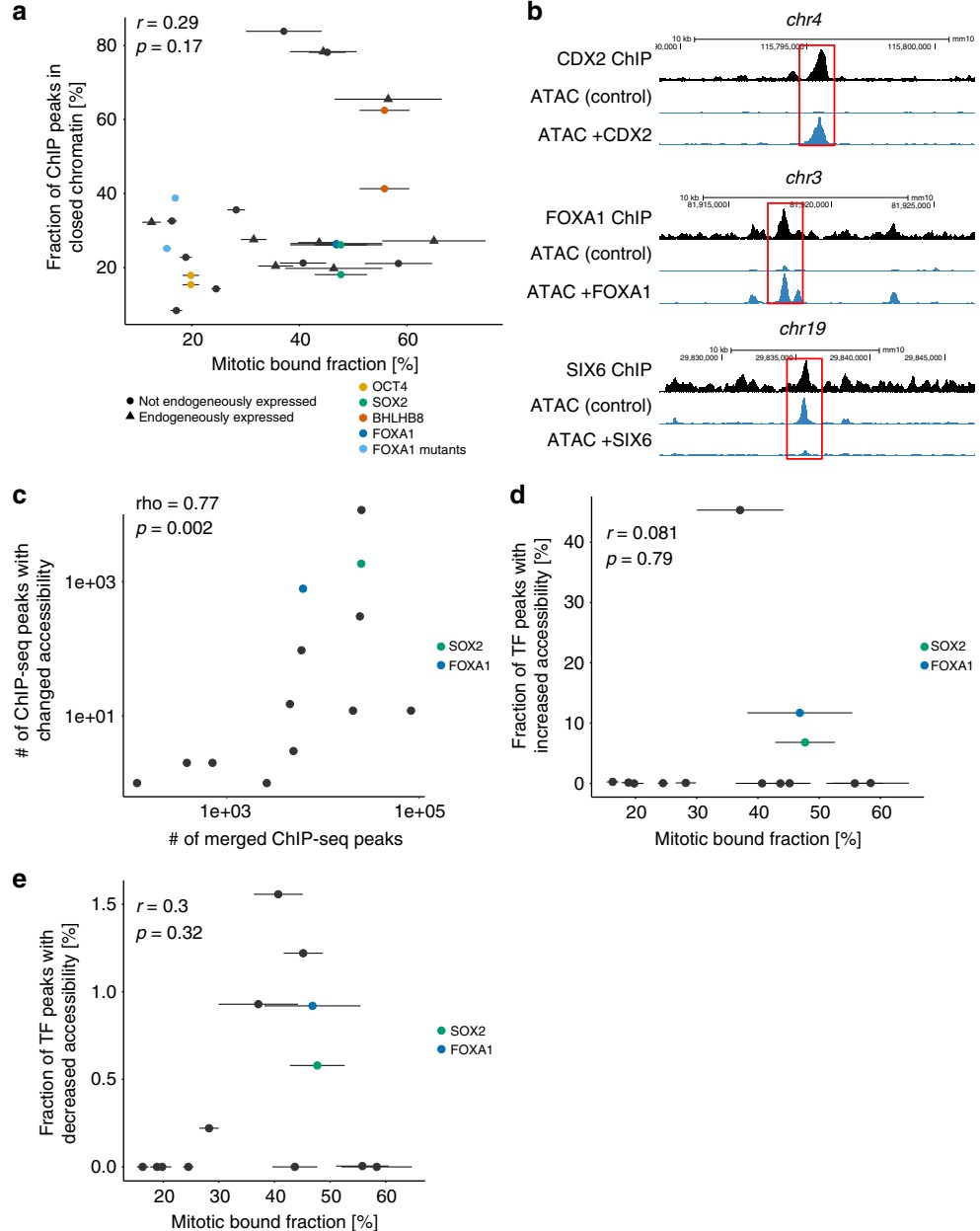

**Fig. 5** TF impact on chromatin accessibility as a function of occupancy and MBF. **a** Correlation between the mitotic bound fraction and the fraction of peaks in closed chromatin regions (devoid of ATAC-seq signal). Duplicates are indicated for OCT4 (yellow), SOX2 (green), BHLHB8 (red), and FOXA1 (dark blue). The two FOXA1 mutants are shown in light blue. Triangles: TFs endogenously expressed in NIH-3T3. Circles: TFs not endogenously expressed in NIH-3T3. *r*-value (*r*) and *p*-value (*p*) are based on Pearson correlation. **b** Example tracks of ChIP-seq signal, ATAC-seq signal in control cells, and ATAC-seq signal in TF overexpression cells for CDX2, FOXA1, and SIX6 for three regions with increased or decreased accessibility after TF overexpression (highlighted in red). The y-axis is scaled equally for ATAC-seq signal in control and TF overexpression within each region. **c** Correlation between the number of ChIP-seq peaks for each TF and the number of TF ChIP-seq peaks with significant change of chromatin accessibility upon overexpression of the TF. Known pioneers are indicated in green (SOX2) and blue (FOXA1). Rho-value (Rho) and *p*-value (*p*) are based on Spearman's rank correlation. **d** Correlation between the mitotic bound fraction and the fraction of regions overlapping a ChIP-seq peak and displaying significant increase in their accessibility upon overexpression of the TF. Same color-coding as **c**. *r*-value (*r*) and *p*-value (*p*) are based on Pearson correlation. **e** Correlation between the mitotic bound fraction and the fraction of regions overlapping a ChIP-seq peak and displaying significant decrease in their accessibility upon overexpression of the TF. Same color-coding as **c**. Rho-value (Rho) and *p*-value (*p*) are based on Spearman's rank correlation. Error bars: SEM

inherent ability to modulate chromatin accessibility. However, the higher number of specific binding sites of TFs with a high MBF correlated with an increased absolute number of bound sites in closed chromatin regions, as well as a broader impact on chromatin accessibility. Therefore, the on-average higher impact on chromatin accessibility of TFs with a high MBF is mediated by their enhanced ability to occupy specific sites rather than an

intrinsically higher ability to alter chromatin accessibility at target sites.

In summary, our findings converge on a model in which non-specific DNA binding properties play a central role in determining TF association to mitotic chromosomes and interphase DNA. We propose that non-specific DNA binding governs TF search efficiency for specific binding sites and thereby has a

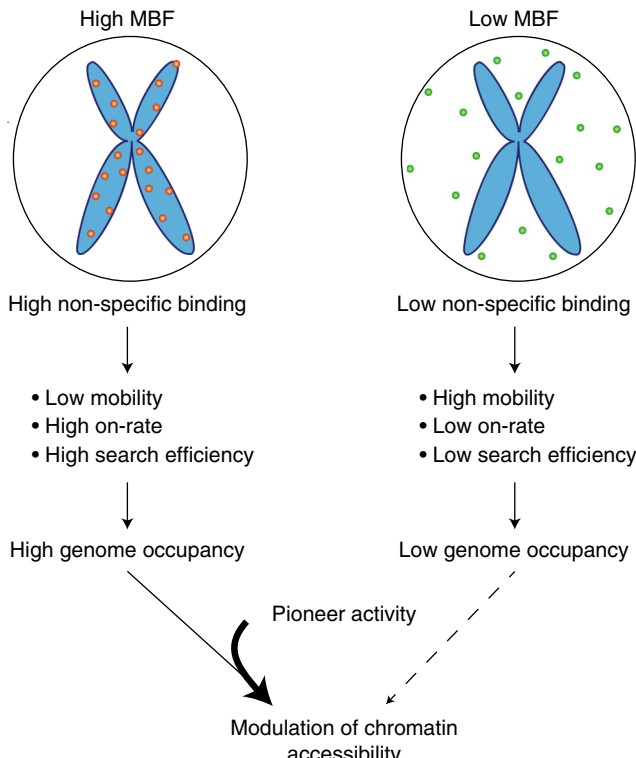

High MBF

High non-specific binding

- Low mobility
- High on-rate
- High search efficiency

High genome occupancy

Low MBF

Low non-specific binding

- High mobility
- Low on-rate
- Low search efficiency

Low genome occupancy

Pioneer activity

Modulation of chromatin accessibility

**Fig. 6** Mitotic chromosome binding predicts TF properties in interphase We propose that the MBF allows to predict TF non-specific DNA binding, which regulates TF mobility, on-rate and search efficiency, thus influencing genome occupancy. Modulation of chromatin accessibility at bound sites depends on genome-wide occupancy but even more so on pioneer activity

modest impact on TF ability to alter the chromatin accessibility landscape (Fig. 6). Future studies should allow further understanding of the molecular basis of differential non-specific DNA binding activity and how this allows to optimize chromatin scanning to find target sites in the genome.

## Methods

**Cell lines and culture.** The E14 mouse embryonic stem (ES) cell line (provided by Didier Trono, EPFL) was used for all experiments involving ES cells. Cells were routinely cultured at 37 °C and 5% $CO_2$ in GMEM (Sigma-Aldrich, #G5154), supplemented with 10% ES cell-qualified fetal bovine serum (ThermoFisher, #16141079), 2 mM sodium pyruvate (Sigma-Aldrich, #113–24–6), 1% non-essential amino acids (ThermoFisher, #11140035), 1% penicillin/streptomycin (BioConcept, #4–01F00H), 2 mM L-glutamine (ThermoFisher, #25030–024), 100 μM 2-mercaptoethanol (Sigma-Aldrich, #63689–25ML-F), leukemia inhibitory factor (LIF, concentration not determined, produced by transient transfection of HEK-293T cells and tested for its potential to maintain pluripotency), 3 μM GSK-3 Inhibitor XVI (Merck Millipore, #361559) and 0.8 μM PD184352 (Sigma-Aldrich, #PZ0181). Cells were grown on 100 mm cell culture dishes coated with 0.1% gelatin, up to confluences of about 70% and split 1:8 to 1:10 every 2–3 days upon trypsinization.

NIH-3T3 cells (provided by Ueli Schibler, University of Geneva) and HEK 293T cells (ATCC) were routinely cultured at 37 °C and 5% $CO_2$ in DMEM (ThermoFisher, #41966029), supplemented with 10% fetal bovine serum (ThermoFisher, #10270106) and 1% penicillin/streptomycin (BioConcept, #4–01F00H). Cells were grown in 100 mm cell culture dishes up to a confluence of 90% and split 1:6 every 3–4 days.

For single molecule imaging experiments, we cultured stable cell lines of NIH3T3 cells with different TFs in DMEM supplemented with 10% FBS (ThermoFisher, #10270106), 1% Sodium Pyruvate (Sigma-Aldrich, #113–24–6), 1% GlutaMax (ThermoFisher, #35050061), 5 μg ml⁻¹ Blasticidin, 2 μg/ml⁻¹ Puromycin and 1% Penicillin-Streptomycin.

**DNA constructs.** A pENTR library containing the coding sequences of 750 mouse transcription factors (TFs) without STOP codon[64] was recombined in the

doxycycline-inducible expression vector described in ref. [32], (hereafter referred as to pLVTRE3G-GW-HA/YPet). This vector allows for dox-inducible expression of each TF fused either to a YPet (a yellow fluorescent protein) or to three HA tags, depending on the presence or absence of Cre recombinase, respectively. Additionally, the coding sequences of Gbx2, Klf2, Klf4, Klf5, Nanog, Tcf3, and Sox17 were added to the library by PCR amplification from cDNA (primers in Supplementary Table 11) extracted from either ES cells maintained in the pluripotent state, or from ES cells differentiated for four days by removal of LIF and 2i. The PCR amplicons were then inserted in the pDONR221 by BP-Gateway recombination (ThermoFisher) and verified by Sanger sequencing. The pENTR library was then recombined into pLVTRE3G-GW-HA/YPet. The resulting lentiviral vector library was used to generate a corresponding ES library for the large-scale quantification of TF binding to mitotic chromosomes and for fluorescence recovery after photobleaching (FRAP) experiments. In the cases of Sox15-YPet and Hoxd10-YPet, the corresponding E14 cell lines did not yield high enough fluorescence intensity signals to perform FRAP experiments. To circumvent this issue, the coding sequences of Sox15-YPet and Hoxd10-YPet were inserted into the pLVTRE3GMCS backbone[32], which allowed obtaining higher expression levels. This was achieved by removing the Oct4 coding sequence from the pLVTRE3G-Oct4-YPet plasmid[32] using SalI and AscI digestion. The coding sequences of Sox15 and Hoxd10 were then PCR-amplified with primers flanked with SalI and AscI sites, digested with these enzymes, and ligated to SalI/AscI-cut pLVTRE3G-Oct4-YPet.

The two FoxA1 non-specific DNA-binding mutants[36] were generated by site-directed mutagenesis before recombination in the pLVTRE3G-GW-HA/YPet, by PCR (primers in Supplementary Table 11) on the FoxA1 pENTR vector.

The constructs for Halo-tagged TFs were generated using restriction cloning, by digesting the pLVTRE3G-GW-HA/YPet with NdeI and AscI to replace the loxP-3xHA-2xSTOP-LoxP-YPet with a NdeI and AscI-cut PCR product of the Halo-Tag coding sequence flanked by NdeI and AscI restriction sites. The selected TFs were then shuttled from the pENTR library into the pLV-TRE3G-GW-Halo by LR-Gateway recombination (ThermoFisher).

The YPet-NLS and YPet-NLS-mutant constructs were generated by amplification of YPet coding sequence from pLVTRE3G-GW-HA/YPet (primers in Supplementary Table 11) to generate the fusion with four different peptides, all bearing five positively charged amino-acids.

The PCR products were inserted in the pLVTRE3GMCS backbone (Deluz et al., 2016) by restriction cloning using SalI and NdeI restriction sites.

The TFs fused to 5 arginine residues were generated by PCR amplification (primers in Supplementary Table 11) of the TF ORF from the corresponding pENTR vectors.

The PCR products were transferred into the pENTR vector by BP-Gateway recombination (ThermoFisher) followed by LR-Gateway recombination (ThermoFisher) into the pLV-TRE3G-GW-HA/YPet.

**Lentivector production and stable cell line generation.** For quantification of mitotic chromosome binding of TFs, lentiviral vector production was carried out by transfection of HEK 293T (cells (ATCC) seeded in 96-well plates with the envelope (PAX2) and packaging (MD2G) constructs together with the lentiviral vector of interest, using the X-tremeGENE 9 DNA Transfection Reagent (Roche, #6365779001). Target E14 cells were engineered to constitutively express rtTA3G-IRES-blasticidin, H2B-mCherry (Addgene, #21217) and Cre recombinase, referred as to E14 ICC for Inducible H2B-mCherry Cre and described in Deluz et al., 2016. E14 ICC cells were then used to generate 757 sub-cell lines (among which we were able to grow 753) allowing dox-inducible expression of each transcription factor. To do so, 4000 cells were seeded in 96-well plates and transduced on two consecutive days with 100 μl of non-concentrated lentiviral vector particles filtered on MultiScreenHTS 0.45 μm filtering plates (Millipore, MSHVS4510). The same transduction protocol was applied to NIH-3T3 ICC cells for quantification of the mitotic bound fraction (MBF) and TF-Hoechst co-localization experiments. Cell lines allowing dox-inducible expression of TF fusions to Halo-Tag for single molecule microscopy, as well as TF fusions to HA tags for ChIP-seq and ATAC-seq experiments were generated by transduction of NIH-3T3 cells constitutively expressing rtTA3G-IRES-blasticidin only.

ES cell lines used for FRAP experiments were generated by transduction of E14 ICC cells with concentrated lentiviral particles generated by calcium phosphate transfection of HEK 293T cells. For all cell lines, selection of transduced cells was performed by addition of the respective antibiotics 48 h after transduction, and antibiotics were maintained in the cell culture medium throughout passaging. For blasticidin selection, we used 8 μg ml⁻¹ (E14 cells) or 5 μg ml⁻¹ (NIH-3T3 cells); for puromycin selection, we used 2 μg ml⁻¹ (all cell lines).

To compare the MBF of endogenously and exogenously expressed SOX2-SNAP fusion proteins (Supplementary Fig. 1b), we used the SOX2-SNAP homozygous knock-in[33] and SOX2-SNAP inducible cell lines[65].

**Live imaging of TFs and MBF quantification.** One day before imaging, cells were seeded in black-walled 96-well plates that were either uncoated (NIH-3T3 cells) or coated with E-Cadherin (R&D systems, #748-EC-050) as previously described[66], and transgene expression was induced with dox (Sigma-Aldrich, #D9891, 500 ng ml⁻¹). Prior to ES cell imaging, the medium was replaced by E14 imaging medium

(FluoroBrite DMEM (ThermoFisher, #A18967–01) supplemented with 10% ES cell-qualified fetal bovine serum (ThermoFisher, #16141079), 2 mM sodium pyruvate (Sigma-Aldrich, #113–24–6), 1% non-essential amino acids (ThermoFisher, #11140035), 1% penicillin/streptomycin (BioConcept, #4–01F00H), 2 mM L-glutamine (ThermoFisher, #25030–024), 100 μM 2-mercaptoethanol (Sigma-Aldrich, #63689–25ML-F), LIF, 3 μM GSK-3 Inhibitor XVI (Merck Millipore, #361559) and 0.8 μM PD18435 (Sigma-Aldrich, #PZ0181)). Prior to NIH-3T3 cell imaging, the medium was replaced by NIH-3T3 imaging medium (FluoroBrite DMEM supplemented with 10% FBS (ThermoFisher, #10270106) and 1 % penicillin/streptomycin). Live fluorescence imaging was performed at the Biomolecular Screening Facility of EPFL on an IN Cell Analyzer 2200 apparatus (GE healthcare) with controlled atmosphere (5% CO2) and temperature (37 °C) and a ×20 magnification objective, using the YFP fluorescence channel for YPet detection, and the TexasRed fluorescence channel for mCherry detection.

For E14 cells, identification of mitotic cells and quantification of the mitotic bound fraction (MBF) were performed using a semi-automated custom pipeline on the CellProfiler software. Briefly, cells in metaphase were automatically discriminated from non-synchronized cells based on shape parameters, and validated by the user. For each confirmed metaphase cell, the selected area was adjusted to precisely define the region containing metaphase chromosomes. This region was then blown-up by 5 pixels and subtracted from a 21 pixel circle drawn around the cell to define the cytoplasmic region. The YPet fluorescence signal was quantified in both regions. For NIH-3T3 cells, regions in the cytoplasm and on the chromosomes were defined by visual inspection. For both cell lines, the MBF was calculated as:

$$MBF = \frac{S_{chrom} \times V_{chrom}}{S_{cyto} \times V_{cyto} \times S_{chrom} \times V_{chrom}} \quad (1)$$

where $S$ is the fluorescence signal and $V$ the average fraction of the volume occupied by either chromosomes or cytoplasm. Vchrom (16%) was determined by confocal microscopy on 32 E14 cells. Briefly, an E14 cell line expressing an H2B-mCherry and a cytoplasmic YFP was seeded in Fluorodishes coated with 5 ng μl$^{-1}$ recombinant mouse E-cadherin Fc chimera protein (R&D systems, #748-EC-050) at a density of 120,000 cells cm$^{-2}$ and fixed after 24 h. Mitotic cells were images in 3D stacks on the LSM700 with z steps of 0.496 μm (otherwise same imaging settings as for the NIH-3T3 imaging). The 3D segmentation of the chromosomes and cytoplasm was done using a pipeline developed by the bioimaging core facility of EPFL on the Imaris software (Bitplane). Vcyto was determined by $1-V_{chrom}$ (84%). For NIH-3T3, volumes measured for E14 cells were used as a proxy. The MBF was averaged over >10 cells per clone for 94% of the TFs and over 4 to 9 cells for 6% (n = 29) of the TFs in E14 cell lines, and over 17 of the 20 NIH-3T3 cell lines, while the 3 others averaged over 4 to 6 cells. For Supplementary Fig. 1g, Supplementary Data 2 from Ginno et al. was downloaded and data for genes present in that study and for which we measured a MBF was merged based on gene name. Mitotic enrichment data was calculated as the log of the mean Chromatome reporter signal in M-phase plus one, divided by the mean Proteome reporter signal in M-phase plus one. For Supplementary Fig. 1h, TFs endogenously expressed in NIH-3T3 were defined as ln (average expression) >2 based on expression data from[54] (GSE66243) and TFs endogenously expressed in ES cells were defined as ln (average E14 2i 72 h expression) >0 based on expression data from[67] (GSE77420).

To compare the MBF between endogenous and exogenous expression of SOX2-SNAP, the cell lines were labeled with 24 nM of the SNAP-Cell_647-SiR dye (New England BioLabs Cat# S9102S) and imaged on the In Cell Analyzer with the Cy5 channel (Supplementary Fig. 1b). MBF were measured manually as described above for NIH-3T3. To compare the MBF between wild type TFs and TFs tagged with 5 arginine residues (Supplementary Fig. 1k), cells were imaged as described above for the large scale quantification. MBF were measured manually as described above for NIH-3T3 cells.

**Immunofluorescence**. Cells were fixed for 30 min with 2% PFA in PBS, permeabilized in 0.5% Triton in PBS, and blocked with PBS and 1% BSA for 30–60 min Samples were incubated with primary antibodies, either anti-HA (anti-HA.11 IgG, BioLegend, # 901501) at 1:500 dilution or anti H3K9me3 (Abcam, #ab8898), at 1:200 dilution in PBS and 1% BSA overnight at 4 °C. Samples were washed twice in PBS. For anti-HA immunostainings, samples were then incubated with either an anti-mouse Alexa 488 antibody (Life Technologies, cat # A21202) or an anti-mouse Alexa 647 antibody (Life Technologies, cat # A-31571) for the OCT4-HA/YPet-Sox2 co-expression experiment (Supplementary Fig.4e). For the anti-H3K9me3 immunostaining, samples were incubated with an anti-rabbit Alexa 647 antibody (Life Technologies, cat # A-31573). All secondary antibodies were used at 1:1000 dilution in PBS and 1% BSA, and left on samples for 45–60 min followed by three washes with 0.1% PBS-Tween, incubation with 2 ng ml$^{-1}$ DAPI, three washes with PBS and 0.1% Tween, and two washes with PBS.

**DNA-binding domain assignment and machine learning analysis**. The number of DNA binding domains per TF and their family were extracted from the UniProt

database. The DBD families (classified as described on http://www.edgar-wingender.de/muTF_classification-1.html, based on ref. [68]) were included in the analysis if present on more than 10 TFs for which we obtained a MBF. TFs with more than one DBD type were included to each of the DBD families. Therefore, in Fig. 1c, some TFs are represented in several boxes. Amino acids were classified as in Lee et al., 2009, into the following categories: positively charged, aromatic, polar, hydrophobic aliphatic, tiny, bulky, and small amino acids (see Supplementary Table 4). Additionally, parameters including the sum of the amino acid grand average of hydropaticity (GRAVY) score of the protein, the total number of consecutive positive amino acids, the total number of consecutive neutral amino acids and the total number of consecutive negative amino acids were calculated as previously described[69]. Most sequence-based parameters were extracted using protr package in R[70]. The sum and the fraction of disordered domains (>5 disordered amino acids in a row) were evaluated using ANCHOR online tool[71]. The dispersion of positive charges was calculated using a sliding window of 5 amino acids to sum the number of arginine and lysine residues and quantified as the variance over the mean of those values. All absolute counts of amino acids were normalized between 0 and 1. Absence or presence of DBD was annotated with 0 and 1, respectively, for each TF. TFs with missing variables were removed from the analysis. The coefficient for each parameter was calculated using a lasso regularized generalized linear model from glmnet package on R[45] on the log of the MBF, and averaged over all the runs. Significant parameters are described as parameters retained by the model in 90% of the runs (n = 500).

**Imaging and co-localization analysis in NIH-3T3 cells**. NIH-3T3 cells (kindly provided by the laboratory of Ueli Schibler, University of Geneva) were seeded in FluoroDishes (WPI, FD35–100) at densities of 36,000 cells cm$^{-2}$ 24 h before imaging and treated with 500 ng ml$^{-1}$ of doxycycline. Shortly before imaging, cells were incubated with 1.62 μM of Hoechst 33342 (Invitrogen, #H3570) for 15 min and washed twice with PBS.

Cells were imaged using a confocal microscope (ZEISS LSM 700 INVERT) with a ×63 objective at 37 °C and 5% CO2. Channel settings were as following: EYFP 2.4% laser power, 700–900 gain, 41.1 μM pinhole; H342 2.6% laser power, 500–700 gain, 39 μM pinhole. Image dimensions: 50.8 × 50.8 μm, 0.05 μm pixel$^{-1}$.

Whole cell signal co-localization was performed by pixel-pixel correlation of the Hoechst and YPet signal images using the R software. Briefly, the Hoechst and YPet images were converted into text images and the correlation score between the two channels was calculated for each cell, using 10 cells per cell line.

The co-localization in different DNA regions was analyzed using an automated image segmentation pipeline in FIJI. Briefly, nuclei were identified and segmented based on the Hoechst signal and 3 regions with high, medium and low Hoechst levels within each nucleus were defined by k-means clustering. Subsequently, the corresponding YPet signal in each of the 3 regions was measured. Ten cells were analyzed per cell line.

**Fluorescence recovery after photobleaching (FRAP)**. E14 cells (kindly provided by the laboratory of Didier Trono, EPFL) were seeded in Fluorodishes coated with 5 ng μl$^{-1}$ recombinant mouse or rat E-cadherin Fc chimera protein (R&D systems, #748-EC-050 and #8144-EC-050 respectively) at densities of 120,000 cells cm$^{-2}$ 24 h before imaging and induced with 500 ng ml$^{-1}$ doxycycline.

Cells were imaged using a confocal microscope (ZEISS LSM 700 INVERT) with a ×63 objective at 37 °C and 5% CO2. Channel settings: EYFP 2.4% laser power, 700–900 gain, 46.2 μm pinhole. Image dimensions: 200 × 200 pixels (19.85 × 19.85 μm).

To image fluorescence recovery after photobleaching, a circular region of interest (ROI) with a diameter of 20 pixels on chromosomes of metaphase cells was selected for bleaching. In addition, two circular control ROIs of the same size were selected, one on the mitotic chromosomes to be used as a non-bleached control and one next to the cell to control for fluctuations of background fluorescence. Cells were first imaged five times with time intervals of 0.38 s to obtain pre-bleach intensity values, and subsequently the selected ROI was bleached for 0.6 s (five iterations) at 100% laser power. Fluorescence recovery was then imaged for 74 s at intervals of 0.38 s.

The same FRAP experiments were performed in interphase cells (one bleached ROI and one non-bleached control ROI within the nucleus, plus one background ROI next to the cell of interest).

To analyze the recovery time, the mean intensities of the bleached ROI and the two non-bleached control ROIs were measured in all time frames. As mitotic chromosomes tended to move throughout the acquisition, the ROIs on mitotic chromosomes were adjusted manually. The recovery curve of the bleached ROI was normalized based on the intensity values before bleaching and on the two control ROIs. The $t_{1/2}$ recovery time was calculated using easyFRAP2mac[72] and averaged over 10 mitotic and 10 interphase cells. To analyze the FRAP data, the intensity was normalized as:

$$I(t) = \left( \frac{\frac{1}{n_{pre}} \sum_{t=1}^{n_{pre}} I(t)_{ROI2'}}{I(t)_{ROI2'}} \right) * \left( \frac{I(t)_{ROI1'}}{\frac{1}{n_{pre}} \sum_{t=1}^{n_{pre}} I(t)_{ROI1'}} \right) \quad (2)$$

Where $n_{pre}$ is the number of pre-bleached frames, $I(t)_{ROI1'}$ is the intensity of the bleached spot minus background and $I(t)_{ROI2'}$ is the intensity of the non-bleached control minus background.

The fit is calculated for each cell as:

$$I_{fit1} = I_0 - a \times e^{-\beta \times t} \qquad (3)$$

The immobile fraction (*Imof*) and $t_{1/2}$ are calculated for each cell as:

$$Imof = 1 - \left( \frac{a}{1 - (I_0 - a)} \right) \qquad (4)$$

$$t_{1/2} = \frac{\ln(2)}{\beta} \qquad (5)$$

**ChIP-seq.** E14 and NIH-3T3 cells were treated overnight with 500 ng ml$^{-1}$ of Doxycycline one day after seeding. Briefly, at least $10^7$ cells were fixed in 1% formaldehyde for 10 min at room temperature, quenched with 250 mM Tris-HCl pH 8.0, washed with PBS, spun down, and stored at $-80\,°C$. The cell pellet was resuspended in 1.5 ml LB1 (50 mM HEPES-KOH pH 7.4, 140 mM NaCl, 1 mM EDTA, 0.5 mM EGTA, 10% Glycerol, 0.5% NP40, 0.25% TritonX-100), incubated 10 min at 4 °C, spun down, and resuspended in 1.5 ml LB2 (10 mM Tris-HCl pH 8.0, 200 mM NaCl, 1 mM EDTA, 0.5 mM EGTA), and incubated 10 min at 4 °C. The pellet was spun down and rinsed twice with SDS shearing buffer (10 mM Tris-HCl pH 8.0, 1 mM EDTA, 0.15% SDS), and finally resuspended in 0.9 ml SDS shearing buffer. All buffers contain 1:100 diluted Protease Inhibitor Cocktail in DMSO (Sigma). The suspension was transferred to a milliTUBE 1 ml AFA fiber and sonicated on a E220 focused ultrasonicator (Covaris) using the following settings: 20 min, 200 cycles, 5% duty, 140W, and input sample aliquots were taken. Sonicated chromatin was incubated with 500 ng of the anti-HA.11 antibody (BioLegend, # 901501) per $10^6$ cells at 4 °C overnight. Protein G Dynabeads (Thermo Fischer) were added to the chromatin and incubated for 3 h at 4 °C. The chromatin was washed several times at 4 °C with 5 min incubation between each wash and 2 min magnetization to collect beads; twice with Low Salt Wash Buffer (10 mM Tris-HCl pH 8.0, 150 mM NaCl, 1 mM EDTA, 1% Triton X-100, 0.15% SDS, 1 mM PMSF), once with High Salt Wash Buffer (10 mM Tris-HCl pH 8.0, 500 mM NaCl, 1 mM EDTA, 1% Triton X-100, 0.15% SDS, 1 mM PMSF), once with LiCl Wash Buffer (10 mM Tris-HCl pH 8.0, 1 mM EDTA, 0.5 mM EGTA, 250 mM LiCl, 1% NP40, 1% sodium deoxycholate, 1 mM PMSF), and finally with TE buffer (10 mM Tris-HCl pH 8.0, 1 mM EDTA, 1 mM PMSF). Beads were finally resuspended in Elution buffer (TE buffer with 1% SDS and 150 mM NaCl), treated with 400 ng/ml Proteinase K and reverse crosslinked at 65 °C 1100 rpm overnight. Input samples were treated with 100 mg/ml RNase A and 400 ng ml$^{-1}$ Proteinase K and reverse crosslinked at 65 °C 1100 rpm overnight. Samples were purified using Qiagen MinElute PCR purification kit. Libraries were prepared with NEB-Next ChIP-seq Library Prep Master Mix Set (NEB, #E6040) using insert size selection of 250 bp. Sequencing was performed using 37 nt paired-end reads on an Illumina NextSeq 500. Reads were aligned to the mouse reference genome mm10 using STAR[73] with settings '--alignMatesGapMax 2000 --alignIntronMax 1 --alignEndsType EndtoEnd -- outFilterMultimapNmax 1'. Duplicate reads were removed with Picard (Broad Institute) and reads not mapping to chromosomes 1–19, X, or Y were removed. For each sample, peaks were called with MACS2[74] with settings '-f BAMPE -g mm' (and '-q 0.01' for Supplementary Fig. 4a). Peaks overlapping peaks called for input (non-immunoprecipitated chromatin) from NIH-3T3 cells and ENCODE blacklisted peaks were discarded[75]. Downsampling of reads (Supplementary Fig. 4c) was done using SAMtools[76]. For HOMER peak calling (Supplementary Fig. 4b), the function findPeaks was used with settings '-style factor' and using Input chromatin as background (for Supplementary Fig. 4b, 1 was added to the number of HOMER-called peaks to avoid zeros). The HOMER2[77] function annotatePeaks.pl was used with settings '-noadj -len 0 -size given' to count the number of reads in peaks and divided by total aligned reads for each sample to get the fraction of reads in peaks. Motif finding was done using the HOMER2 function findMotifsGenome.pl with settings '-size given'. Top motifs were selected as the most significant de novo hit in either the entire peak set for each factor, or in the peaks overlapping (open) or not overlapping (closed) open chromatin based on ATAC-seq peaks in NIH-3T3 (see Supplementary Table 9). Published motifs were selected by taking the most enriched motif that corresponded to each factor or its TF family (from either de novo or known motif search) (see Supplementary Table 9). For those factors where no published motif was enriched, JASPAR-annotated motifs were used where possible (DLX1, DLX6, HLF, SIX6, and TEAD1)[78]. In the final analysis, only those motifs with a p-value lower than 0.05 were kept, and factors with top motifs corresponding to SeqBias were discarded. The HOMER2 function scanMotifGenomeWide.pl was used to calculate the occurrence of motifs. For Fig. 4a–b, the average number of ChIP-seq peaks were used for duplicated factors (BHLHB8, FOXA1, SOX2). bigWig files were generated by the deepTools function bamCoverage (with setting '--normalizeUsingRPKM'). Genome tracks were visualized in the UCSC genome browser.

**ATAC-seq.** NIH-3T3 cells were plated and treated with 500 ng ml$^{-1}$ of Doxycycline 48 h before the experiment. ATAC-seq experiments were performed using in-house prepared Tn5 transposase (in-house production[79]). Briefly, $5*10^4$ cells were pelleted and washed with 1× ice cold PBS at 800×$g$ for 5 min Cells were resuspended in 50 μl of ice-cold ATAC lysis buffer (10 mM Tris-HCl pH 7.4, 10 mM NaCl, 3 mM MgCl2, 0.1% NP40), and pelleted at 800×$g$ for 10 min at 4 °C. Cells were subsequently resuspended in 50 μl of transposition reaction mix containing 0.5 μM of Tn5 transposase in TAPS-DMF buffer (10 mM TAPS-NaOH, 5 mM Mgcl2, 10% DMF) and incubated at 37 °C for 30 min The transposed DNA was purified using a DNA purification kit (Zymo Research #D4003) and eluted in 12 μl of water. A 65 μl PCR reaction was setup with 10 μl of transposed DNA, 0.5 μM of forward primer Ad1_noMX, 0.5 μM of multiplexing reverse primer Ad2.x[80], 0.6x SYBR® Green I, and 1× PCR Master Mix (NEB #M0544). The samples were thermocycled at 72 °C for 5 min, 98 °C for 30 s, followed by 5 cycles at 98 °C for 10 s, 63 °C for 30 s and 72 °C for 1 min A 15 μl aliquot was analyzed by qPCR to determine the number of additional cycles needed to avoid amplification saturation. The amplified ATAC libraries were purified using a DNA purification kit (Zymo Research #D4003) and size selected using Agencourt AMPure beads (0.55× unbound fraction followed by 1.2× bound fraction). All libraries were sequenced with 75-nucleotide read length paired-end sequencing on a Illumina NextSeq 500 with 30–60 million reads being sequenced for each sample.

Two replicates were performed for each TF overexpression sample, and four replicates for control cells expressing only rtTA3G. Sequencing and read alignment was performed as described above for ChIP-seq. To determine regions that were accessible in the NIH-3T3 genome, we performed one ATAC-seq replicate of the parental NIH-3T3 cell line and called peaks as described above for ChIP-seq. ChIP-seq peaks from all TFs analyzed and ATAC-seq peaks from all samples were merged using BEDTools[81] into two separate files and the number of ATAC-seq reads in these peak sets was calculated for each sample using HOMER2 as described above for ChIP-seq. Read counts were normalized with edgeR[82] using TMM. The limma[83] package was used to call regions that had differentially abundant ATAC-seq reads between the TF overexpression condition and control cells expressing only rtTA3G, with an adjusted p-value of 0.05 as cutoff. For plots displayed in log scale, 1 was added to the number of regions with affected accessibility for each TF. Note that the number of TF peaks used in the ATAC-seq data represents the number of merged peaks for ChIP-seq duplicated factors. bigWig files were generated by merging replicate bam files with SAMtools followed by the deepTools functions bamCoverage (with setting '--normalizeUsingRPKM'). log2-ratio bigWig files were generated using the deepTools function bamCompare based on merged replicates for each TF overexpression sample over merged replicates for control (rtTA3G). Heatmaps were generated using deepTools computeMatrix (with setting 'reference-point') and plotHeatmap centered around the merged peak set for each TF.

**Single molecule fluorescence microscopy.** Cells were seeded on glass-bottom dishes (Delta T culture dishes, Bioptechs, Pennsylvania, USA) one day before the measurement. To induce the expression of Halo-tagged TFs, 10 ng ml$^{-1}$ doxycycline was added 4 h after seeding the cells. Before imaging, Halo-tagged TFs were labeled with fluorescent SiR ligand (kindly provided by Kai Johnsson, EPFL, Switzerland, (Promega, #G8251)) according to the HaloTag protocol (Promega). DNA was labeled with 0.3 μg ml$^{-1}$ Hoechst 33342 (ThermoFisher, #62249). Single molecule imaging was performed in phenol free Opti-MEM (ThermoFisher, #11058021) at 37 °C for up to 120 min

Single molecule microscopy was performed on a custom built microscope described previously[84]. Briefly, light of a 405 nm laser (Laser MLD, 200 mW, Cobolt, Solna, Sweden) and a 638 nm laser (IBEAM-SMART-640-S, 150 mW, Toptica, Gräfelfing, Germany) were collimated, combined using a dichroic mirror, controlled by an AOTF (AOTFnC-400.650-TN, AA Optoelectronics, Orsay, France) and used for inclined illumination in a fluorescence microscope (TiE, Nikon, Tokyo, Japan) with a high-NA objective (100×, NA 1.45, Nikon, Tokyo, Japan). Fluorescent light was filtered by a multiband emission filter (F72–866, AHF, Tübingen, Germany) and detected by an EMCCD camera (iXon Ultra DU 897U, Andor, Belfast, UK).

To investigate binding properties of TFs, we used two different illumination schemes: (i) continuous movies of cells illuminated with the 638 nm laser to excite SiR were recorded with 50 ms camera integration time, preceded and followed by a snapshot of cells illuminated with the 405 nm laser to excite Hoechst 33342 (50 ms integration time). (ii) movies were recorded in which snapshots of cells illuminated for 50 ms with the 638 nm laser were alternated every 550 ms with snapshots of cells illuminated for 50 ms with the 405 nm laser (1.2 s total cycle time).

We detected Halo-TF molecules based on their fluorescence intensity above the background level and determined their position using a 2D Gaussian fit[84]. Halo-TF molecules were identified as bound to chromatin when they were detected within a spherical region of 160 nm of diameter for 2 consecutive frames (i.e., for at least 100 ms in illumination scheme (i) and for at least 1.2 s in illumination scheme (ii)).

We separated chromosomal regions into three classes, bright, intermediate, and dark, according to their Hoechst 33342 intensity by means of two user defined intensity thresholds. Subsequent analysis steps were performed separately for each intensity class. We assigned bound molecules to a chromosomal region by comparing their position of first appearance with the preceding Hoechst image.

This allowed accounting for cellular movements during long acquisition times in illumination scheme (ii).

We determined the fraction of bound TF-Halo molecules per area by dividing the number of molecules identified as bound in illumination scheme (i), $N_{bi}$, by the area $A_a$ of the respective Hoechst intensity class, $a$ = bright, intermediate or dark. We then divided the result by the total number of detected Halo-TF molecules, $N_{tot}$, and the video capture time $T$. The result is a measure of binding frequency, which we referred to as pseudo on-rate ($\psi$on-rate):

$$\psi\text{on}_i = \frac{N_{bi}}{A_a N_{tot} T} \tag{6}$$

The $\psi$on-rate of long bound Halo-TF molecules was determined analogously using the respective molecule counts of illumination scheme ii:

$$\psi\text{on}_{ii} = \frac{N_{bii}}{A_a N_{tot} T} \tag{7}$$

To obtain a simple measure for the residence time $\tau_{res}$ of specifically bound molecules, we calculated the average time that Halo-TF molecules spent bound in illumination scheme (ii):

$$\tau_{res} = \frac{1}{N_{tot}} \sum_{j=1}^{N_{tot}} t_{bii,j} \tag{8}$$

where $t_{bii,j}$ is the binding time of molecule $j$ in illumination scheme (ii). We then ordered the TFs with respect to this measure.

**Quantification and statistical analysis**. Statistical analyses of violin plots of TF distributions (Fig. 1e), on the MBF for NLS mutants (Supplementary Fig. 1j) and on the MBF for TFs tagged with five arginine residues (Supplementary Fig. 1k) were performed using Wilcoxon rank-sum test. For plots with linear scales, $r$-values and $p$-values are based on Pearson correlation (Figs. 2d-g, 3c-k, 4a-b, 5a, 5d-e, and Supplementary Figs. 1a, 1d, 1f, 2a-b, 3d-f). For plots displayed in log scale, Rho-values and $p$-values are based on Spearman's rank correlation (Figs. 4d-e, 5c, Supplementary Figures 4a-d, 4h-j, 5a and 5c-g). Note that correlations were calculated using averages for replicates. Duplicate values were not averaged for the FOM of top motifs due to enrichment for different motifs.

**Code availability**. The code to analyze residence times and pseudo-on-rates from single molecule tracks is available as Matlab script on the Dryad Data Repository (https://doi.org/10.5061/dryad.9pc7458). Other codes used in this study are available from the corresponding author upon reasonable request.

**Reporting summary**. Further information on experimental design is available in the Nature Research Reporting Summary linked to this article.

## Data availability
Data supporting the findings of this manuscript are available from the corresponding author upon reasonable request. A reporting summary for this Article is available as a Supplementary Information file. ChIP-seq and ATAC-seq data that support the findings of this study have been deposited in GEO (Gene Expression Omnibus) with the accession code GSE119784.

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

## Acknowledgements

This work was supported by the Teofilo Rossi di Montelera e di Premuda Foundation advised by CARIGEST SA, and an anonymous donor advised by CARIGEST SA (D.M.S), the German Research Foundation [GE 2631/1–1 to J.C.M.G.], the European Research Council (ERC) under the European Union's Horizon 2020 Research and Innovation Programme [637987 ChromArch to J.C.M.G.] and the DFG Graduate School of Molecular Medicine at Ulm University (to H.A.). We thank Bart Deplancke for providing the pENTR transcription factor library, Rubina Davtyan and Katharina Soukup for their assistance in analyzing parts of the single molecule data, Vincent Gardeux for his assistance in computing the machine learning algorithm, Bastien Mangeat and Elisa Cora from the EPFL Gene Expression Core Facility (EPFL-GECF) for high throughput sequencing, Fabien Kuttler from the EPFL Biomolecular Screening Facility (EPFL-BSF) and José Artacho from the EPFL Bioimaging and Optics Core Facility (EPFL-BIOP) for assistance in imaging, and Philipp Bucher for critical reading of the manuscript.

## Author contributions

Conceptualization, M.R. and D.M.S.; Methodology, M.R., D.M.S., H.A., and J.C.M.G.; Software, M.R., E.T.F., and T.K.; Formal Analysis, M.R., A.B.A., E.T.F., and T.K; Investigation, M.R., A.B.A., H.A., and C.D.; Resources, D.M.S. and J.C.M.G., Writing–Original Draft, M.R., D.M.S., and J.C.M.G.-Reviewed manuscript, M.R., E.T.F., D.M.S., and J.C.M.G.; Funding Acquisition, D.M.S., and J.C.M.G.; Supervision, D.M.S. and J.C.M.G.

## Additional information

**Competing interests:** The authors declare no competing interests.

