## [Peer Review File · Nature Communications]

Reviewers' Comments:

Reviewer #1:

Remarks to the Author:

In this manuscript, M. Raccaud and colleagues provide the most comprehensive analysis so far reported of TF behavior in mitotic cells using live imaging. They then correlate the ability of a TF to interact with the mitotic chromosomes with several aspects, from intrinsic features of the proteins, to the biophysical and molecular parameters of interphase binding for a subset of TFs. The paper is certainly interesting and represents a valuable resource for follow up studies. However, it is rather correlative and does not bring any major conclusion to move the field forward. A more specialized journal would be more appropriate; however, I suggest the authors to consider the points below to improve the manuscript.

1/ In the introduction, the authors have a specific paragraph on the current knowledge of mitotic binding by TFs. My understanding of the field suggests that they are taking for granted a number of yet unproved concepts, as listed below. The authors should be more careful in the introduction and, also, fully comment on these key issues in their discussion.

1a/ They claim mitotic binding is rather independent of sequence-specific interactions. I do not think this has yet been proven, in particular given the difficulties that have been several times reported to crosslink TFs to mitotic chromatin. Nevertheless, ATAC-seq assays published by the Tjian group suggests many TFs bind to mitotic chromatin rather specifically (PMID: 27855781). TFs may bind in a very dynamic way in mitosis, as suggested in the aforementioned publication.

1b/ The authors previously published that Sox2 is a factor binding mitotic chromatin but with a very small number of site-specific interactions. However, in an independent report from the Apostolou lab (PMID: 28514649), Sox2 was shown to establish a large amount of site-specific interactions.

1c/ My understanding from the few mutational analyses so far published is that altering sequence-specific binding does significantly alter the ability of a TF to interact with the mitotic chromatin, in contrast to what the authors claim (PMID: 27723719 and PMID: 23355396).

2/ As mentioned above, PFA has been shown to alter the mitotic behavior of some TFs. Now that the authors have an impressive system to perform live imaging of 500+ TFs, I think it would be mandatory to assess how many of the TFs they tested are subject to the PFA artefact. It is possible that from this analysis, they will be able to increase their correlative power to the intrinsic properties they correlate to mitotic binding.

3/ The authors use interphase ChIP-seq of a few TFs to correlate site-specific binding to their mitotic behavior. My impression is that this paper could represent a milestone in the field if the mitotic binding profiles of a large subset of TFs was provided. I strongly encourage the authors to reconsider their work and include mitotic ChIPseq for as many TFs as possible – at least to those analyzed in 3T3 and by FRAP/SMT.

4/ Technical points:

4a/ In some cases, the authors have analyzed only 10 mitotic cells (from a single experiment?) – is this really sufficient? I am concerned by the statistical power of their studies

4b/ How did the authors select the ORF of each TF? In many cases several isoforms do likely exist and they may have drastically different properties.

4c/ Do the authors have any idea of the functionality of the fusion proteins they use? Can they rule out neomorphic properties arising from the overexpression?

4d/ The authors should provide snapshots of TF binding and accessibility, as well as provide a more canonical representation of genome-wide metrics (metaplots; statistical assessment of ATAC-seq changes, etc)

Reviewer #2:

Remarks to the Author:

Many studies measure biophysical properties in either a small number of TFs or in-vitro conditions. The work of Raccaud et al. is a unique attempt to extensively dissect fundamental biophysical properties of a large number of transcription factors in-vivo. Of special interest are the correlation between the on-rates and the number of ChIP-seq peaks and the connections between non-specific binding and facilitated search process. To quantitatively characterize the nuclear diffusion and binding dynamics of transcription factors is essential to understand the regulation of transcription. Therefore, the work of Raccaud et al. may attract a lot of attention in the scientific community.

Comments

- 1) To ensure the paper is accessible to a broad community of readers the authors should state better what are the basic parameters that dictate in theory the diffusion and binding dynamics of transcription factors at the beginning as well as how their measurements relate, either directly or indirectly, to these parameters throughout the text.
- 2) One of the major hypotheses in the paper is that mitotic binding is mainly the result of non-specific interactions. If that's true one would expect that the MBF would be independent of the total concentration of TFs for a large range of values. The authors could show whether this is true by calculating the MBF at different TF concentrations either using the intrinsic cell-to-cell variability or using different dox concentrations.
- 3) Where are TFs previously reported to be highly enriched on mitotic chromosome and present in our library, such as FOXA1, GATA1, GATA4, SOX2, RUNX2, ESRRB, RBPJ and HNF1b on Figure 1B? They should be highlighted? Why is CDX2 shown? Where are the borders of the three bins in Figure 1B?
- 4) On Page 4 the authors write: "The absolute charge per DBD was the most distinctive parameter between TFs enriched on mitotic chromosome (dark blue) versus those that are not (medium and light blue), suggesting that electrostatic interactions play an important role in mitotic chromosome association of TF". This suggestion should be experimentally verified.
- 5) The section "TF mobility does not depend on TF size or residence time on specific sites" is confusing, perhaps because its counterintuitive results. At the beginning the authors state that FRAP recovery time depends on 3D diffusion, specific and non-specific binding but the result indicates that is the on-rate the key parameter.
- 6) The concentration of the TFs and therefore the overall expression level of the TFs is a very important parameter, which could affect the measured MBFs. Is the MBF of the exogenous fusion proteins comparable to the endogenous TF levels? The authors should carry out experiments testing whether the MBF of an endogenous TF is comparable to its exogenously expressed counterpart?
- 7) On page 9 of the results, the authors selected 13 TFs to analyse chromatin accessibility which have no endogenous counterparts in the cells. Also, in the Discussion section on page 10, the authors stress that most of the TFs studied are not endogenously expressed in NIH-3T3 cells. Are MBFs influenced by the fact that a factor is expressed in a given cell type or expressed in an environment where the TF has no function (not expressed)?

Minor Comments:

1. The authors mentioned in the Introduction co-localization studies using fluorescence microscopy

to study the association of TFs with mitotic DNA (page 2). However, it has been shown that this approach can lead to false negative results concerning the binding of TFs to mitotic DNA, especially if chemical fixation protocols and immunofluorescence were used to detect the TFs (Teves et al., 2016). The authors should comment this finding in the Introduction.

2. The authors state in the first part of the Results section that they could not detect any differences between the MBFs of some chosen TFs in ES and NIH-3T3 cells which would indicate a large cell-type independence (page 3). This statement is too strong as only two cell-types were tested so far and only a subset of the TFs were used for the comparison. Therefore, this sentence should be rewritten.

3. To test the mobility of TFs during mitosis and interphase, the authors selected 15 out of the 38 previously used TFs in NIH-3T3 cells (page 5). How were these 15 TFs selected? Why were not all 38 TFs tested? The authors should indicate how they selected the TFs for the mobility measurements.

4. In Figure 5B-D only SOX2 and FOXA1 are indicated in the legend. Why are the authors not showing the names of all TFs?

5. There is a mistake in the legend of supplemental Table 5. In the description line for the SEM_Pixel_Corr_HoechstYPet is a space missing between "the pixel" and the word "correlation" is written twice in a row.

Reviewer #3:

Remarks to the Author:

A number of transcription factors have been observed to bind to mitotic chromosomes. In some cases, it is proposed that this is a mitotic bookmarking mechanism that regulates programs of transcription in daughter cells as they come out of mitosis and reassemble nuclei. This manuscript profiles a large number of transcription factors, characterizes their mitotic chromosome binding, and identifies tendencies for binding characteristics. This is an important subject. The unique contribution in this manuscript is the look at so many transcription factors in side-by-side experiments.

Specific Comments:

1) The differential localizations of transcription factors in Fig 1 is worthy of comment, especially the two factors that appear to be highly concentrated in peri-centric heterochromatin.

2) Throughout the manuscript the authors say they measure "mobility" as an insight into factor function. In fact they measure "binding". There is no experiment they report that is likely to measure free diffusion of transcription factors. The frequent discussion of "mobility" is therefore likely to confuse readers. Transcription factors bind with different degrees of specificity and diffuse between binding events. They are measuring the former and not the later.

3) Binding is first measured here by the fraction bound to chromosomes in imaging experiments and then by FRAP. The FRAP presentation requires some specification of the method of analysis, compete with equation, and not just stating the mac program used. There are methods of calculation that would give an immobile fraction of tightly bound molecules that do not exchange over the time of the experiments. Given the model presented of non-specific binding leading to specific binding, this would be important information. Sample, FRAP recovery curves should be presented and table six should probably present immobile fractions along with $t_{1/2}$ s. Presenting a few sample image series of recovery would also be reassuring. FRAPing mitotic chromosomes is not easy since they are moving and squirming. Finally, for FRAP, I did not see the number of cells FRAPed for each transcription factor. $n=?$ should be a part of the FRAP table.

3) The interpretation of single cell experiments establishing "residence time" were not clearly explained. This probably also requires a specification of the equations, a more complete

description of the method, and a more convincing discussion of the calculation especially why they think they are measuring on-rate constants and not off-rate constants.

4) One big issue in the model presented is that it may be confounded by cooperating factors that may mediate the chromosome binding of screened transcription factors. The authors deal with this for Oct4, but the issue probably has a broader significance in the interpretation of the screens and for the model and this could be better discussed.

5) Figure 6, the model, is not helpful. It does not depict a link between interphase binding (left) and mitotic binding (right). A more informative cartoon might be useful.

6) Some discussion of the functions of chromosome binding might be helpful, about whether the tendencies reported shed light on proposed "bookmarking" mechanisms.

7) A little realism would help. The experiments are heroic in looking at so many factors in parallel but they identified tendencies and not rules that will predict the behavior of the $n+1$ th transcription factors. So, phrases like "remarkable predictive value" are probably not appropriate.

8) There is a literature of single cell analyses of transcriptional regulator binding to chromosomes, such as to integrated chromosomal arrays, that is relevant and probably should be considered and cited.

Reviewers' comments:

Reviewer #1 (Remarks to the Author):

In this manuscript, M. Raccaud and colleagues provide the most comprehensive analysis so far reported of TF behavior in mitotic cells using live imaging. They then correlate the ability of a TF to interact with the mitotic chromosomes with several aspects, from intrinsic features of the proteins, to the biophysical and molecular parameters of interphase binding for a subset of TFs. The paper is certainly interesting and represents a valuable resource for follow up studies. However, it is rather correlative and does not bring any major conclusion to move the field forward. A more specialized journal would be more appropriate;

While we agree with the reviewer about the central role played by the correlations we made between different parameters, these allowed revealing deep connections between dynamic parameters of TFs. We thus strongly disagree that our study “does not bring any major conclusion to move the field forward”. As stated by reviewer 2, “*Of special interest are the correlation between the on-rates and the number of ChIP-seq peaks and the connections between non-specific binding and facilitated search process*”. We believe that our study is unique in this respect and paves the way for further investigation of the links between non-specific binding properties and TF search efficiency.

However, I suggest the authors to consider the points below to improve the manuscript.

1/ In the introduction, the authors have a specific paragraph on the current knowledge of mitotic binding by TFs. My understanding of the field suggests that they are taking for granted a number of yet unproved concepts, as listed below. The authors should be more careful in the introduction and, also, fully comment on these key issues in their discussion.

1a/ They claim mitotic binding is rather independent of sequence-specific interactions. I do not think this has yet been proven, in particular given the difficulties that have been several times reported to crosslink TFs to mitotic chromatin. Nevertheless, ATAC-seq assays published by the Tjian group suggests many TFs bind to mitotic chromatin rather specifically (PMID: 27855781). TFs may bind in a very dynamic way in mitosis, as suggested in the aforementioned publication.

We thank the reviewer for this comment and would like to provide some clarifications. It is true that transcription factors associating with mitotic chromosomes have been shown to display not only non-specific but also specific DNA binding interactions, by several groups including ours (e.g. Kadauke et al., Cell 2012, Caravaca et al., Genes & Development 2013, Teves et al., eLife 2016, Deluz et al., Genes & Development 2016). Indeed the Tjian group (Teves et al., eLife 2016) has shown evidence for Sox2 footprints on mitotic chromosomes, arguing for the presence of specific DNA binding events. However, our point is that the bulk of the association *that is observed by fluorescence microscopy* is mostly reflecting non-specific DNA interactions. We have now further clarified this in the manuscript (see revised abstract and page 3 of the revised manuscript). In the discussion section we also further clarify the distinction between mitotic bookmarking, which implies specific DNA binding,

from mitotic chromosome association as observed by fluorescence microscopy (page 12 of the revised manuscript).

1b/ The authors previously published that Sox2 is a factor binding mitotic chromatin but with a very small number of site-specific interactions. However, in an independent report from the Apostolou lab (PMID: 28514649), Sox2 was shown to establish a large amount of site-specific interactions.

We thank the reviewer for bringing up this issue. As correctly stated by the reviewer, there is a large discrepancy in the ChIP-seq findings between different laboratories. Of note, a recent preprint from the Navarro laboratory (Festuccia et al., bioRxiv 2018) dissects the impact of different fixation methods on mitotic ChIP-seq results and reaches conclusions very similar to ours concerning the very few mitotic ChIP-seq peaks for Sox2, and strongly contrasting with those obtained by Liu et al. Nevertheless, the reasons for these discrepancies are still poorly understood, highlighting the lack of a consensus methodology to obtain reliable and reproducible mitotic ChIP-seq results (see discussion on page 12 of the revised manuscript, and discussion on technical issues associated with mitotic ChIP-seq in our reply to point 2/).

1c/ My understanding from the few mutational analyses so far published is that altering sequence-specific binding does significantly alter the ability of a TF to interact with the mitotic chromatin, in contrast to what the authors claim (PMID: 27723719 and PMID: 23355396).

We thank the reviewer for raising this point. We do actually not exclude that specific DNA binding can contribute to mitotic chromosome binding, and we would like to reiterate that we are referring to the bulk of mitotic chromosome association that is observed by fluorescence microscopy. We had actually cited these two studies as ref 31 and 35 of the following citation: “the mild or null sensitivity [of co-localization of TFs with mitotic chromosomes that is observed by microscopy] to alterations of specific DNA binding properties^{31,35}” (page 3 of the revised manuscript). We believe this statement is accurate, because:

- In PMID 27723719, the specific DNA binding mutant of ESRRB indeed leads to a mild reduction of mitotic chromosome binding (Fig.5b). In fact, this mutant was generated by the replacement of 3 amino acids by glycine residues, including the suppression of two positively charged amino acids. As we show in our revised manuscript, positive charges in the DNA binding domain have a strong impact on mitotic chromosome association (revised Supplementary Fig.1j-k). Thus, in addition to suppressing specific DNA interactions, this may lead to decreased non-specific DNA interactions, which were not assessed in their study. Furthermore, the authors did not generate non-specific binding mutants to test the contribution of non-specific DNA binding of ESRRB to mitotic chromosome binding.
- In PMID 23355396, the sequence-specific binding mutant of FoxA1 was actually shown to display unchanged association to mitotic chromosomes, in contrast to

the non-specific DNA binding that displayed strongly reduced mitotic chromosome association (Fig.5B of PMID 23355396). Thus, this study in fact provides strong evidence for the central role of non-specific rather than specific binding in mediating FoxA1 association with mitotic chromosomes as observed by fluorescence microscopy.

2/ As mentioned above, PFA has been shown to alter the mitotic behavior of some TFs. Now that the authors have an impressive system to perform live imaging of 500+ TFs, I think it would be mandatory to assess how many of the TFs they tested are subject to the PFA artefact. It is possible that from this analysis, they will be able to increase their correlative power to the intrinsic properties they correlate to mitotic binding.

We first would like to clarify that in our study, we did not use PFA fixation at all to study mitotic chromosome association of TFs, but relied on live cell imaging of TFs fused to the YPet fluorescent protein in unsynchronized cell populations. Indeed, PFA has been shown to disrupt mitotic chromosome association as visualized by fluorescence microscopy for at least 13 different transcription factors analyzed by 4 different laboratories (9 shown in Teves et al., eLife 2016, 2 in Pallier et al., Molecular Biology of the Cell 2003, 1 in Lerner et al., Nucleic Acids Research 2016, and 1 in Kumar et al., Biochimica et biophysica acta 2008). Therefore, the existence of this artifact is already well established. It seems to us that further studying the PFA artifact will thus not represent a major advance in the field and is out of the scope of the present study.

3/ The authors use interphase ChIP-seq of a few TFs to correlate site-specific binding to their mitotic behavior. My impression is that this paper could represent a milestone in the field if the mitotic binding profiles of a large subset of TFs was provided. I strongly encourage the authors to reconsider their work and include mitotic ChIPseq for as many TFs as possible – at least to those analyzed in 3T3 and by FRAP/SMT.

We thank the reviewer for bringing up this issue. Importantly, there is currently no consensus method on how to perform mitotic ChIP-seq. As the reviewer pointed out, very different results were obtained by different groups for the same transcription factors (Sox2 by Liu et al., 2017 versus Deluz et al. 2016 and Festuccia et al., 2018, as well as Oct4 in Liu et al., 2017 versus Festuccia et al., 2018). While this might be due to several reasons such as the mode of fixation (PFA versus DSG, see Deluz et al 2016, Liu et al. 2017 versus Festuccia et al. 2018), as well as different ways to purify mitotic cells (nocodazole synchronization followed by purification of H3S10p population (Kadauke et al. 2012, Deluz et al. 2016, versus nocodazole synchronization followed by mitotic shake-off (Festuccia et al. 2016 and 2018, Liu et al. 2017), this essentially remains an unsolved issue. Therefore the extremely laborious task of performing mitotic ChIP-seq experiments for dozens of TFs is unlikely to lead to meaningful and reliable results. Furthermore, we think that this is largely out of the scope of our study since it is not related to the claims we make in our manuscript.

4/ Technical points:

4a/ In some cases, the authors have analyzed only 10 mitotic cells (from a single experiment?) – is this really sufficient? I am concerned by the statistical power of their studies

We privileged measurements of live, unsynchronized cells at the metaphase stage to avoid any artifact in our measurements due to differences in mitotic stages analyzed or to drug synchronization. As a consequence, this limits the number of cells that can be analyzed. Nevertheless, for 88 TFs the MBF was measured from 2 independent biological experiments, for 42 TFs from 3 independent experiments, and for 3 TFs from 4 independent experiments. The MBF obtained from the different biological replicates were similar. Furthermore, the robustness of our measurements is shown by 1) the strong correlation with confocal microscopy measurements (revised Supplementary Fig.1c, d) and measurements performed in NIH-3T3 cells (revised Supplementary Fig.1e, f); 2) the fact that previously identified mitotic chromosome binding TFs also display a high MBF in our experiments. We would also like to point out that we do not make claims about statistically significant differences between the MBFs of individual TFs.

4b/ How did the authors select the ORF of each TF? In many cases several isoforms do likely exist and they may have drastically different properties.

We used a library that has been previously developed by the Deplancke laboratory (PMID: 23917988), with the addition of a few TFs as described in the methods section of our manuscript. While we fully agree that different TF isoforms may have different properties, we did not aim at comprehensively describing mitotic chromosome association of all possible mouse TFs and their different isoforms, but rather to quantify a set of TFs that is large enough to understand how mitotic chromosome binding informs us on other TF properties in interphase.

Note that while we were verifying ORFs of the TF library, we found one TF (HMGB1) for which the sequence was truncated. We thus re-run all analyses that included data from HMGB1 in the previous version of the manuscript and revised the corresponding figures accordingly.

4c/ Do the authors have any idea of the functionality of the fusion proteins they use?

We assume that the reviewer refers to DNA binding/mitotic chromosome association when referring to function, since this is what we examine in our study, irrespective of the function of TFs in regulating gene expression. We can indeed not fully rule out that in some cases, the YPet tag disturbs mitotic chromosome association. However, we so far never observed that the position or nature of the fluorescent tag significantly perturbs mitotic chromosome binding. In our study (Deluz et al., *Genes & Development* 2016), we show that N or C-terminal fusions of Sox2 and Oct4 to Ypet or their fusion to Firefly luciferase behave similarly. In that study we also showed that Nanog, Esrrb or Klf4 behave similarly when fused to YPet or a SNAP tag. Finally, our results are also fully consistent with those from Teves et al., *eLife* 2016 for all TFs analyzed by both studies (5 in total). Therefore, while we cannot fully exclude that some TFs are altered in their mitotic chromosome association by

fusion to YPet, we think this is extremely unlikely to affect our conclusions, which are drawn from a large number of TFs. In fact, such perturbations are more likely to add noise to (and thereby artificially weaken) the correlations we observed. Finally, our ChIP-seq analysis was performed on TFs tagged with 3 HA tags, adding only 53 amino acids to the C-terminus, and we were able to retrieve the published DNA binding motif for a large number of them even though most TFs were not expressed in their endogenous context.

Can they rule out neomorphic properties arising from the overexpression?

We thank the reviewer for raising this point. We have now performed extensive new analysis showing that mitotic bound fractions are generally not correlated to overexpression levels (revised Supplementary Fig. 1a). Concerning other neomorphic properties such as non-physiological alteration of gene expression due to overexpression, these are not relevant to our study.

4d/ The authors should provide snapshots of TF binding and accessibility, as well as provide a more canonical representation of genome-wide metrics (metaplots; statistical assessment of ATAC-seq changes, etc)

While we had included genome tracks of TF binding in Supplementary Fig. 4a of the first submission, these are now displayed in the main figures (revised Fig. 4c). We have now also included three genome track examples of binding sites that are altered in their accessibility after TF overexpression (revised Fig. 5b). In addition, we now show heatmap metaplots allowing to visualize the overall changes in ATAC-seq accessibility for all assessed factors, enabling an overview of the ChIP signal and log₂-fold change between TF overexpression and control (revised Supplementary Fig. 5b). The statistical assessment of the changed loci in the ATAC-seq data was done using edgeR and limma. We have now clarified this point in the main text (page 11 of the revised manuscript) and in the Methods section (page 21 of the revised manuscript).

Reviewer #2 (Remarks to the Author):

Many studies measure biophysical properties in either a small number of TFs or in-vitro conditions. The work of Raccaud et al. is a unique attempt to extensively dissect fundamental biophysical properties of a large number of transcription factors in-vivo. Of special interest are the correlation between the on-rates and the number of ChIP-seq peaks and the connections between non-specific binding and facilitated search process. To quantitatively characterize the nuclear diffusion and binding dynamics of transcription factors is essential to understand the regulation of transcription. Therefore, the work of Raccaud et al. may attract a lot of attention in the scientific community.

We thank the reviewer for the appreciation of our work.

Comments

1) To ensure the paper is accessible to a broad community of readers the authors should

state better what are the basic parameters that dictate in theory the diffusion and binding dynamics of transcription factors at the beginning as well as how their measurements relate, either directly or indirectly, to these parameters throughout the text.

We thank the reviewer for pointing this out, and we agree that we should have described the theory underlying TF dynamics more thoroughly. We now more extensively discuss this in the introduction section of the revised manuscript (page 2 of the revised manuscript).

2) One of the major hypotheses in the paper is that mitotic binding is mainly the result of non-specific interactions. If that's true one would expect that the MBF would be independent of the total concentration of TFs for a large range of values. The authors could show whether this is true by calculating the MBF at different TF concentrations either using the intrinsic cell-to-cell variability or using different dox concentrations.

This is an excellent point. We have now analyzed the correlations between the MBF and overexpression levels using the intrinsic cell-to-cell variability in TF overexpression levels, for 21 different TFs spanning a broad range of MBF, and for which we measured the MBF in at least 19 cells. Overall, these parameters did not display a consistent correlation (revised Supplementary Fig.1a). Thus, the MBF does not decrease at high TF overexpression levels, suggesting that TFs do not "saturate" mitotic chromatin. This is thus compatible with non-specific DNA interactions as being mainly responsible for the association of TFs with mitotic chromosomes as observed by fluorescence microscopy.

3) Where are TFs previously reported to be highly enriched on mitotic chromosome and present in our library, such as FOXA1, GATA1, GATA4, SOX2, RUNX2, ESRRB, RBPJ and HNF1b on Figure 1B? They should be highlighted? Why is CDX2 shown? Where are the borders of the three bins in Figure 1B?

We thank the reviewer for this very good point, which actually allowed us to spot a mistake that we made. We inadvertently used a list of TFs well-known to bind mitotic chromosomes independently of those we measured in our experiments, and this list was swapped with the one corresponding to TFs that we also measured in our experiments. For FOXA1, GATA1, SOX2 and ESRRB, we had indeed quantified their MBF; in contrast, GATA4, HNF1b and RUNX2 were not part of the library, and RBPJ was one of the library TF that we were unable to quantify because of low transduction efficiency. Furthermore, we had omitted HMGB2 and HMGN1, which are known to bind mitotic chromosomes and that we have quantified in our experiments. We have now highlighted FOXA1, GATA1, SOX2, ESRRB, HMGB2 and HMGN1 in Figure 1b and corrected the corresponding results section (page 4 of the revised manuscript).

CDX2 was chosen randomly to illustrate a TF that is enriched on mitotic chromatin, but we have now removed the corresponding microscopy image from Fig.1b and show SOX2 instead. We have also highlighted the borders of the three bins shown in Fig.1b.

4) On Page 4 the authors write: "The absolute charge per DBD was the most distinctive parameter between TFs enriched on mitotic chromosome (dark blue) versus those that are

not (medium and light blue), suggesting that electrostatic interactions play an important role in mitotic chromosome association of TF". This suggestion should be experimentally verified.

We had actually performed experiments showing that adding positive charges to YPet increases its colocalization with mitotic chromosomes (now revised Supplementary Fig.1j). We have now performed further experiments by adding five positive charges to four different TFs (3 with a low and 1 with an intermediate MBF), resulting in an increase of their MBF (revised Supplementary Fig.1k). Note that we avoided targeting the positive charges within or directly next to the DNA binding domain, since this entailed a high risk of perturbing its structure and thereby its DNA binding properties.

5) The section "TF mobility does not depend on TF size or residence time on specific sites" is confusing, perhaps because its counterintuitive results. At the beginning the authors state that FRAP recovery time depends on 3D diffusion, specific and non-specific binding but the result indicates that is the on-rate the key parameter.

We thank the reviewer for raising this point, and we agree that this section was not clear enough. Indeed, the FRAP recovery time depends on 3D diffusion, specific and non-specific DNA binding, but here we intended to dissect which of these parameters is explaining the *differences* in FRAP recovery time *between* TFs. We have now clarified this on page 7 of the revised manuscript.

6) The concentration of the TFs and therefore the overall expression level of the TFs is a very important parameter, which could affect the measured MBFs. Is the MBF of the exogenous fusion proteins comparable to the endogenous TF levels? The authors should carry out experiments testing whether the MBF of an endogenous TF is comparable to its exogenously expressed counterpart?

We thank the reviewer for raising this point. To satisfy the reviewer's request, we have now further investigated the MBF of a Sox2-SNAP homozygous knock-in cell line and compared it to a Sox2-SNAP overexpression cell line, and we found comparable values for both (revised Supplementary Fig.1b).

7) On page 9 of the results, the authors selected 13 TFs to analyse chromatin accessibility which have no endogenous counterparts in the cells. Also, in the Discussion section on page 10, the authors stress that most of the TFs studied are not endogenously expressed in NIH-3T3 cells. Are MBFs influenced by the fact that a factor is expressed in a given cell type or expressed in an environment where the TF has no function (not expressed)?

We thank the reviewer for these important points. We reasoned that endogenous expression of TFs may mask the impact of their overexpressed counterparts on chromatin accessibility. We thus on purpose selected TFs that have no endogenously expressed counterparts for ATAC-seq experiments, thus asking ask how *de novo expression* of these alters chromatin accessibility.

To determine whether MBFs are influenced by its endogenous expression, we have now

compared the MBF for TFs that are endogenously expressed versus those that are not, both for ES cells and NIH-3T3 cells, and found no differences between these two groups (revised Supplementary Fig. 1h).

Minor Comments:

1. The authors mentioned in the Introduction co-localization studies using fluorescence microscopy to study the association of TFs with mitotic DNA (page 2). However, it has been shown that this approach can lead to false negative results concerning the binding of TFs to mitotic DNA, especially if chemical fixation protocols and immunofluorescence were used to detect the TFs (Teves et al., 2016). The authors should comment this finding in the Introduction.

We thank the reviewer for raising this point – indeed artifacts from chemical fixation were one of the reasons why we chose to use a live cell imaging approach for our study. We have now specified in the text that we were referring to the use of fluorescent fusions to study mitotic chromosome association of TFs, and we also mention problems arising from chemical fixation (page 3 of the revised manuscript).

2. The authors state in the first part of the Results section that they could not detect any differences between the MBFs of some chosen TFs in ES and NIH-3T3 cells which would indicate a large cell-type independence (page 3). This statement is too strong as only two cell-types were tested so far and only a subset of the TFs were used for the comparison. Therefore, this sentence should be rewritten.

We fully agree that this was an overstatement – we have now rewritten this sentence to make this statement less strong (page 4 of the revised manuscript).

3. To test the mobility of TFs during mitosis and interphase, the authors selected 15 out of the 38 previously used TFs in NIH-3T3 cells (page 5). How were these 15 TFs selected? Why were not all 38 TFs tested? The authors should indicate how they selected the TFs for the mobility measurements.

We thank the reviewer for raising this point. We did not test 38 TFs since some of these were expressed at relatively low levels upon dox induction, thus making FRAP measurements less reliable. Thus we selected 18 TFs (15 for both mitotic and interphase FRAP, 3 with a very low MBF allowing only for interphase FRAP) based on: 1) their expression levels upon dox induction being high enough to allow reliable FRAP measurements; 2) their span of a large range of MBF. We have now better specified this in the text (page 7 of the revised manuscript).

4. In Figure 5B-D only SOX2 and FOXA1 are indicated in the legend. Why are the authors not showing the names of all TFs?

In Figure 5b-d (now revised Fig. 5c-e) we show how exogenous TF expression impacts local chromatin accessibility. We have chosen to highlight only SOX2 and FOXA1 as these are well-established pioneer TFs, which can thus easily be compared to other TFs that were

considerably less potent (with the exception of CDX2) in modifying chromatin accessibility. The values for all other TFs can be found in Table S10.

5. There is a mistake in the legend of supplemental Table 5. In the description line for the SEM_Pixel_Corr_HoechstYPet is a space missing between “the pixel” and the word “correlation” is written twice in a row.

We thank the reviewer for spotting this mistake, which has now been corrected.

Reviewer #3 (Remarks to the Author):

A number of transcription factors have been observed to bind to mitotic chromosomes. In some cases, it is proposed that this is a mitotic bookmarking mechanism that regulates programs of transcription in daughter cells as they come out of mitosis and reassemble nuclei. This manuscript profiles a large number of transcription factors, characterizes their mitotic chromosome binding, and identifies tendencies for binding characteristics. This is an important subject. The unique contribution in this manuscript is the look at so many transcription factors in side-by-side experiments.

We thank the reviewer for her/his appreciation of our study.

Specific Comments:

1) The differential localizations of transcription factors in Fig 1 is worthy of comment, especially the two factors that appear to be highly concentrated in peri-centric heterochromatin.

We thank the reviewer for raising this interesting point. Indeed, pericentric DNA was shown to localize to DNA-dense Hoechst regions observed in nuclei of fibroblasts and other differentiated cell types. However we prefer not to make any claim about the nature of these heterochromatin regions since we did not directly identify pericentric DNA in our study. We have now commented on this on page 6 of the revised manuscript.

2) Throughout the manuscript the authors say they measure “mobility” as an insight into factor function. In fact they measure “binding”. There is no experiment they report that is likely to measure free diffusion of transcription factors. The frequent discussion of “mobility” is therefore likely to confuse readers. Transcription factors bind with different degrees of specificity and diffuse between binding events. They are measuring the former and not the later.

We thank the reviewer for bringing up this important point that deserves to be clarified. While we indeed do not directly measure 3D diffusion, FRAP measures the exchange of bleached and unbleached molecules, which depends not only on non-specific and specific DNA binding as stated by the reviewer, but also on 3D diffusion that can significantly contribute to FRAP recovery times (see Mazza D et al., Nucleic Acids Research 2012).

Therefore, determining the origin in differences of FRAP recovery times between TFs requires to consider possible differences in 3D diffusion. What we show in the manuscript is that potential differences in 3D diffusion between TFs do not explain differences in FRAP recovery $t_{1/2}$.

We thus also believe that using the term “mobility” when referring to FRAP recovery $t_{1/2}$ is appropriate, as is commonly done in the literature (see for example PMIDs 11389456, 15695095, 19234451).

3) Binding is first measured here by the fraction bound to chromosomes in imaging experiments and then by FRAP. The FRAP presentation requires some specification of the method of analysis, compete with equation, and not just stating the mac program used. There are methods of calculation that would give an immobile fraction of tightly bound molecules that do not exchange over the time of the experiments. Given the model presented of non-specific binding leading to specific binding, this would be important information. Sample, FRAP recovery curves should be presented and table six should probably present immobile fractions along with $t_{1/2}$ s. Presenting a few sample image series of recovery would also be reassuring. FRAPing mitotic chromosomes is not easy since they are moving and squirming. Finally, for FRAP, I did not see the number of cells FRAPed for each transcription factor. $n=?$ should be a part of the FRAP table.

We thank the reviewer for raising these important points. We have now computed the immobile fractions for each TF. We have also included more details on the method and the equations used (page 20 of the revised manuscript) to determine the $t_{1/2}$ recovery (Equation 5) and the immobile fractions (Equation 4). We also now display examples of FRAP microscopy time-series in revised Fig.3a, b.

We found the immobile fractions to be correlated with the MBF (revised Figure 3f, g and Supplementary Table 6), in line with “non-specific binding leading to specific DNA binding”. Concerning the challenge of FRAP on mitotic chromosomes, we think this is a rather minor issue since their movement occurs on substantially slower time scales than the FRAP recovery. In fact, FRAP on mitotic chromosomes has been successfully performed before by many groups, including us (Caravaca et al., Genes & Dev 2013, Festuccia et al., Nature Cell Biology 2016 & 2018, Deluz et al., Genes & Dev 2016, Teves et al., eLife 2016). We now also mention the number of FRAPed cells in the figure legend and in Supplementary table 6.

3) The interpretation of single cell experiments establishing “residence time” were not clearly explained. This probably also requires a specification of the equations, a more complete description of the method, and a more convincing discussion of the calculation especially why they think they are measuring on-rate constants and not off-rate constants.

We thank the reviewer for bringing this up, and we agree that we have not been sufficiently clear here. We are actually not claiming to measure “on-rate constants and not off-rate constants”. We calculated two different parameters from the single molecule tracking data: the frequency of binding events, which we call pseudo on-rate (Equations 6 and 7, page 23 of the revised manuscript), and the average residence time (Equation 8, page 23 of the revised manuscript). The frequency of binding events is a measure of events per time and corresponds to an on-rate. The inverse of the residence time would be equal to the off-rate.

As suggested we now improved the description of single molecule experiments in the main text (page 8 of the revised manuscript) and the methods section (page 23 of the revised manuscript).

4) One big issue in the model presented is that it may be confounded by cooperating factors that may mediate the chromosome binding of screened transcription factors. The authors deal with this for Oct4, but the issue probably has a broader significance in the interpretation of the screens and for the model and this could be better discussed.

We thank the reviewer for raising this important point. Indeed Oct4 occupancy in interphase is increased by the presence of Sox2, and in an earlier study from our laboratory we actually found that Sox2 expression increases mitotic chromosome association of Oct4 (Deluz et al., Genes & Dev 2016). Thus it is indeed possible that some TFs interact with mitotic chromosomes partly or completely through cooperativity, and that the same mechanism also affects their search efficiency in the interphase genome. We have now included this in the discussion (page 12 of the revised manuscript). Importantly, this does not impact the general principle that TFs that are more prone to mitotic chromosome association are more efficient at finding specific target sites in the genome, but extends this notion to indirect interactions and those depending on cooperativity.

5) Figure 6, the model, is not helpful. It does not depict a link between interphase binding (left) and mitotic binding (right). A more informative cartoon might be useful.

This point of criticism is well-taken. We now provide a more detailed model to better illustrate how mitotic chromosome binding reflects DNA-binding properties of TFs in interphase (revised Fig.6).

6) Some discussion of the functions of chromosome binding might be helpful, about whether the tendencies reported shed light on proposed “bookmarking” mechanisms.

This is also a good point. We have now added a paragraph to discuss how our observations are related to mitotic bookmarking (page 12 of the revised manuscript).

7) A little realism would help. The experiments are heroic in looking at so many factors in parallel but they identified tendencies and not rules that will predict the behavior of the n+1th transcription factors. So, phrases like “remarkable predictive value” are probably not appropriate.

We have now toned down this statement (page 12 of the revised manuscript), as well as on page 3 where we replaced “is predictive of” by “correlates with”, and also toned down the titles of the 3rd and 7th subsections in the results part (page 6 and 9 of the revised manuscript, respectively).

8) There is a literature of single cell analyses of transcriptional regulator binding to chromosomes, such as to integrated chromosomal arrays, that is relevant and probably should be considered and cited.

This is indeed an interesting comparison to make. An important distinction with the observation of mitotic chromosome association is that array approaches are designed for the observation of specific DNA binding events for a given TF species. We now have mentioned these studies in the introduction (page 3 of the revised manuscript).

Reviewers' Comments:

Reviewer #1:

Remarks to the Author:

The authors have tried to carefully reply to my concerns. Unfortunately, I am not convinced by their claims. I think it is very important to clarify that I do not see in the paper any evidence indicating that the global coating of the mitotic chromosomes they observe is necessarily reflecting non-specific interactions with the DNA. TFs can surely interact non-specifically with different components of the mitotic chromosomes beyond DNA. Just to give two possibilities, among many others, positively charged TFs may engage in electrostatic interactions with the highly phosphorylated H3 histones characterising mitotic chromatin; also, they could be somehow trapped in the chromosomal periphery, as it was already proposed by the Earnshaw lab. The fact that several labs have already shown nuclear localisation signals to be required for the chromosomal retention of TFs, also suggests that the bulk of the interactions are not necessarily based on non-specific DNA interactions.

If the authors cannot prove that the bulk of the chromosomal signal represents non-specific binding to DNA, then the basis of the correlations they make with the capacity of these TFs to search for specific binding sites in interphase loses all its potential strength. Therefore, I still believe the paper is far too correlative and speculative to warrant publication in Nature Communications. As I suggested, this paper clearly requires, in my opinion, a molecular assessment of TF binding to DNA in mitotic cells. According to several reports, this is perfectly doable even to identify non-specific interactions, as the Zaret group showed several years ago.

In conclusion, I sincerely appreciate the very ambitious nature of this paper; it represents the major attempt so far executed to thoroughly assess TF binding in mitosis. However, the correlations they made are based on an unproven assumption that may have negative consequences in the future, should it turn out to be untrue.

Reviewer #2:

Remarks to the Author:

The authors have satisfactorily answered my concerns.

Reviewer #3:

Remarks to the Author:

This improved manuscript profiles a large selection of transcription factors, characterizing their mitotic chromosome binding, and identifies tendencies for binding specificity and affinity. While this is correlative, the manuscript makes a significant contribution by profiling many transcription factors in side-by-side experiments. The manuscript does not much address whether this binding at mitosis has epigenetic or regulatory significance, or whether mitotic binding just reflects the fact that transcription factors have non-specific and specific binding to DNA and chromosomes are DNA. More discussion of this would have been appreciated but it would have gone beyond the data. It is the global nature of the profiling make this a significant report.

The authors have been responsive to the first review and made changes that improve the experimental presentation. The presentation and discussion about FRAP has been corrected and improved. I would still contend that they are measuring rate limiting binding and unbinding steps and not diffusion. As pointed out, diffusion scales inversely with the radius of a globular protein. However, radius scales with the third root of the mass, so proteins need to be hugely massive for a diffusion equation to have relevance here. This is a small point in this manuscript and should not

affect publication. They have recalculated FRAP experiments in a way quantifying a tightly bound fraction that does not exchange during the experiment, something that may correlate with specific vs non-specific binding, and they show FRAP recovery curve examples.

Figure 6, a presentation of the final model, has been improved. Equations for calculations of residence times have been added. Appropriate text changes make the context of these experiments clearer.

REVIEWERS' COMMENTS:

Reviewer #1 (Remarks to the Author):

The authors have tried to carefully reply to my concerns. Unfortunately, I am not convinced by their claims. I think it is very important to clarify that I do not see in the paper any evidence indicating that the global coating of the mitotic chromosomes they observe is necessarily reflecting non-specific interactions with the DNA. TFs can surely interact non-specifically with different components of the mitotic chromosomes beyond DNA. Just to give two possibilities, among many others, positively charged TFs may engage in electrostatic interactions with the highly phosphorylated H3 histones characterising mitotic chromatin; also, they could be somehow trapped in the chromosomal periphery, as it was already proposed by the Earnshaw lab. The fact that several labs have already shown nuclear localisation signals to be required for the chromosomal retention of TFs, also suggests that the bulk of the interactions are not necessarily based on non-specific DNA interactions.

If the authors cannot prove that the bulk of the chromosomal signal represents non-specific binding to DNA, then the basis of the correlations they make with the capacity of these TFs to search for specific binding sites in interphase loses all its potential strength. Therefore, I still believe the paper is far too correlative and speculative to warrant publication in Nature Communications.

Here we provide some clarifications. First, earlier studies already provided strong evidence that the bulk of mitotic chromosome association is mediated by nonspecific DNA binding, as we have explained in detail in the Introduction section. Second, while we cannot exclude that TFs could also interact with highly phosphorylated H3 histones or other mitotic-specific chromatin/chromatin periphery components, the fact the TFs associated with mitotic chromosomes also display increased colocalization with DNA in interphase suggests that DNA interactions play a direct role in mitotic chromosome association. Thus, while we agree that other mechanisms could also contribute to the decoration of mitotic chromosomes by TFs, there is strong converging evidence that non-specific DNA binding plays a major role in these interactions.

Furthermore, in our manuscript, we then show that i) mitotic chromosome binding is largely insensitive to overexpression levels (Revised Supplementary Figure 1a), and thus this is not in line with specific DNA binding as mediating mitotic chromosome association ; ii) Differences in TF diffusion (Revised Supplementary Figure 2) or DNA residence times (Revised Figure 3j,k) do not explain differences in TF mobility, thus strongly suggesting that these are due to differences in non-specific DNA binding which thereby increase TF on-rates.

Concerning the statement made by the reviewer on nuclear localization signals : the previous studies investigating the impact of NLS on mitotic chromosome association did not allow distinguishing between an active transport mechanism and electrostatic, non-specific TF-DNA interactions. In our manuscript we have shown that an equivalent number of positive charges impact mitotic chromosome binding even more than NLS do (Revised Supplementary Figures 1j) and adding positive charges to TFs increases their MBF (Revised Supplementary Figures 1k). This shows that the effect of NLS on mitotic chromosome association is mediated by its positive charges.

As I suggested, this paper clearly requires, in my opinion, a molecular assessment of TF binding to DNA in mitotic cells. According to several reports, this is perfectly doable even to identify non-specific interactions, as the Zaret group showed several years ago.

We disagree. As we already explained in our previous reply, mitotic ChIP-seq is plagued by highly discrepant results between labs, and thus further studies are needed to clarify how to perform these experiments in a reliable and reproducible manner.

In conclusion, I sincerely appreciate the very ambitious nature of this paper; it represents the major attempt so far executed to thoroughly assess TF binding in mitosis. However, the correlations they made are based on an unproven assumption that may have negative consequences in the future, should it turn out to be untrue.

We disagree. This present manuscript together with earlier studies have provided convincing evidence that mitotic chromosome association as visualized by fluorescence microscopy is a proxy for non-specific DNA binding properties. The finding that these properties are of predictive values for TF search efficiency in interphase will be of major interest to the community. Future studies will be helpful in better characterizing these non-specific interactions in live cells, but these will require the development of new methods to visualize these highly transient events.

Reviewer #2 (Remarks to the Author):

The authors have satisfactorily answered my concerns.

We thank the reviewer for his positive assessment of our revised manuscript.

Reviewer #3 (Remarks to the Author):

This improved manuscript profiles a large selection of transcription factors, characterizing their mitotic chromosome binding, and identifies tendencies for binding specificity and affinity. While this is correlative, the manuscript makes a significant contribution by profiling many transcription factors in side-by-side experiments. The manuscript does not much address whether this binding at mitosis has epigenetic or regulatory significance, or whether mitotic binding just reflects the fact that transcription factors have non-specific and specific binding to DNA and chromosomes are DNA. More discussion of this would have been appreciated but it would have gone beyond the data. It is the global nature of the profiling make this a significant report.

We thank the reviewer for his positive assessment of our revised manuscript.

The authors have been responsive to the first review and made changes that improve the experimental presentation. The presentation and discussion about FRAP has been corrected and improved. I would still contend that they are measuring rate limiting binding and unbinding steps and not diffusion. As pointed out, diffusion scales inversely with the radius of a globular protein. However, radius scales with the third root of the mass, so proteins need to be hugely massive for a diffusion equation to have relevance here. This is a small point in this manuscript and should not affect publication.

We thank the reviewer for this point on which we agree. We have now better specified this in the manuscript and rephrased the corresponding part :

Even though differences in TF radius are predicted to be very small since they scale with the third root of their mass, we assessed the correlation between TF-YPet molecular weight and FRAP $t_{1/2}$ recovery (Supplementary Fig. 2a, b and Supplementary Table 6).

They have recalculated FRAP experiments in a way quantifying a tightly bound fraction that does not exchange during the experiment, something that may correlate with specific vs non-specific binding, and they show FRAP recovery curve examples.

Figure 6, a presentation of the final model, has been improved. Equations for calculations of residence times have been added. Appropriate text changes make the context of these experiments clearer.

We thank the reviewer for his positive assessment of our revisions.